# LumiTex: Towards High-Fidelity PBR Texture Generation with Illumination Context

**Jingzhi Bao**[1*], **Hongze Chen**[2*], **Lingting Zhu**[3], **Chenyu Liu**[4], **Keyang Luo**[5], **Runze Zhang**[5],
**Zeyu Hu**[5], **Yingda Yin**[5], **Weikai Chen**[5], **Xin Wang**[5], **Zehong Lin**[6†], **Jun Zhang**[2], **Xiaoguang Han**[1,7,8†]

[1]School of Science and Engineering, The Chinese University of Hong Kong, Shenzhen
[2]The Hong Kong University of Science and Technology
[3]The University of Hong Kong,    [4]Peking University,    [5]LIGHTSPEED,    [6]Lingnan University
[7]Shenzhen Future Network of Intelligence Institute
[8]Guangdong Provincial Key Laboratory of Future Networks of Intelligence, The Chinese University of Hong Kong, Shenzhen
zehonglin@ln.edu.hk, hanxiaoguang@cuhk.edu.cn

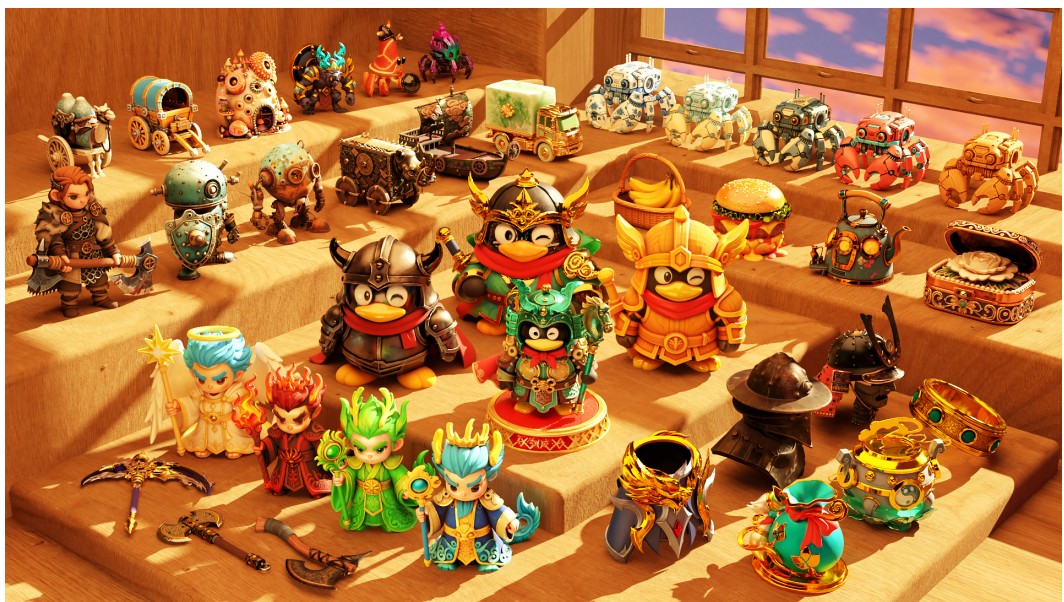

Figure 1. A collection of textured meshes with PBR materials generated by LumiTex, capturing both intricate details and convincing physical realism across diverse object categories.

## Abstract

Physically-based rendering (PBR) provides a principled standard for realistic material–lighting interactions in computer graphics. Despite recent advances in generating PBR textures, existing methods fail to address two fundamental challenges: 1) materials decomposition from image prompts under limited illumination cues, and 2) seamless and view-consistent texture completion. To this end, we propose LumiTex, an end-to-end framework that comprises three key components: (1) a multi-branch generation scheme that disentangles albedo and metallic–roughness under shared illumination priors for robust material understanding, (2) a lighting-aware material attention mechanism that injects illumination context into the decoding process for physically grounded generation of albedo, metallic, and roughness maps, and (3) a geometry-guided inpainting module based on a large view synthesis model that enriches texture coverage and ensures seamless, view-consistent UV completion. Extensive experiments demonstrate that LumiTex achieves state-of-the-art performance in texture quality, surpassing both existing open-source and commercial methods. Project page: https://lumitex.vercel.app.

---

*Equal contribution.
†Corresponding authors.

# 1 INTRODUCTION

Physically-based rendering (PBR) is the industry standard for material and lighting representation in games, films, and AR/VR. PBR textures encode key surface properties such as albedo and metallic–roughness (MR), allowing for realistic visual interactions under diverse lighting conditions. Nevertheless, it is challenging to generate PBR textures, which requires both accurate physical characterization of materials and consistency across multiple viewpoints.

To obtain the high-quality texture from a mesh and image, multi-view texturing has emerged as the dominant framework across both research (Huang et al., 2024c; Liang et al., 2025b; Zhang et al., 2024a; Zhao et al., 2025) and commercial systems (TripoAI, 2025; Meshy, 2025; Team, 2025b). At the core of this framework are multi-view diffusion models, which jointly reason about the underlying 3D structure and consistent appearance across all views through multi-view attention (Shi et al., 2023). This mechanism links pixels across viewpoints to ensure coherent geometry and appearance, after which a back-projection step aggregates the synthesized views into UV texture maps. A key strength of this paradigm is that it inherits the visual quality and diversity of pre-trained image diffusion models. Consequently, adapting these models to handle PBR-aware multi-view generation provides a unified pipeline that naturally bridges reference images with textured 3D assets.

Recent advances in PBR texture generation, primarily driven by the remarkable capabilities of multi-view diffusion models, can be broadly categorized into two main approaches. The first line of research (Zhang et al., 2024c; Zhu et al., 2025; Hong et al., 2024; Munkberg et al., 2025) adopts a two-stage approach: it first generates multi-view images that encode baked environment lighting (shaded images), and then obtains PBR textures either through optimization or by employing dedicated multi-view inference models, e.g., IDArb (Li et al., 2025b). These shaded images supply rich illumination cues that are crucial for accurate material decomposition. However, optimization-based methods such as DreamMat (Zhang et al., 2024c) are severely limited by long optimization time. While multi-view inference models can help in efficiency, existing methods like MuMA (Zhu et al., 2025) often produce inferior materials, as the generated intermediates are frequently suboptimal and poorly aligned with the training inputs, as illustrated in Fig. 2(a).

The second stream (Zhao et al., 2025; He et al., 2025; Zhu et al., 2024; Huang et al., 2025) focuses on jointly generating multi-view albedo, metallic, and roughness images with multi-channel diffusion (see Fig. 2(b)) and subsequently baking multi-view images to the UV space via camera projection (Chen et al., 2023b; Richardson et al., 2023). However, these methods suffer from three key limitations. Firstly, generating accurate multi-view material images is challenging since image diffusion models (Ho et al., 2020; Podell et al., 2023; Lin et al., 2024; Esser et al., 2024) lack sufficient material priors, and reference images provide only

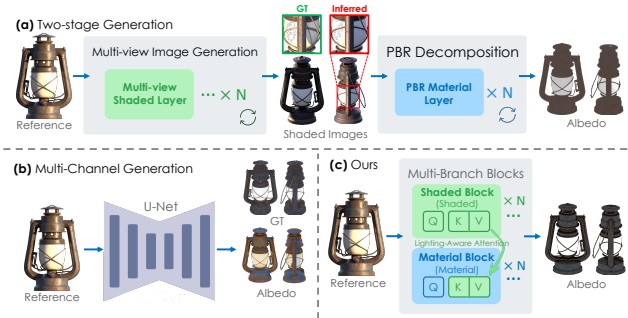

Figure 2. **Illustration of Different PBR Modeling.** Unlike (a) two-stage PBR texture generation with suboptimal intermediates, and (b) end-to-end approach w/o multi-view shaded features, (c) our multi-branch design leverages multi-view consistent lighting features and surpasses the generation quality of other models.

limited illumination cues. Secondly, the domain gap between albedo and metallic-roughness (MR) is often overlooked. Albedo reflects the intrinsic surface color, whereas MR encodes illumination-dependent material properties. However, multi-channel approaches typically predict them jointly within a shared output space. This entangled representation, without explicit contextual guidance, hinders accurate material decomposition and compromises the physical plausibility of the generated textures. Lastly, existing datasets suffer from severe data imbalance. While shaded and albedo images are relatively abundant, high-quality metallic and roughness maps are much more scarce, limiting effective supervision for PBR materials.

To address these limitations, we propose LumiTex, an end-to-end framework that jointly generates multi-view shaded images and PBR maps. Our key idea is to integrate *multi-stage inference* into

a single one to overcome the reliance on imperfect shaded intermediates while alleviating the data imbalance and retaining rich illumination cues via *multi-stage training*. Specifically, we first train a multi-view illumination context branch to reconstruct shaded images across views, capturing rich and consistent lighting cues as an explicit illumination context. To address the domain gap between albedo and MR in joint modeling, we then introduce a lighting-aware material attention mechanism in the material branches that separately guide albedo and MR by the illumination context for channel-specific reasoning. By combining the generative prior of diffusion models with the rich illumination context extended from reference images, LumiTex enables high-fidelity and physically plausible texture generation in a fully end-to-end manner.

To further enhance the quality and completeness of generated textures, we introduce a general texture inpainting strategy based on a large view synthesis model (LVSM) (Jin et al., 2025). Specifically, we train a geometry-guided LVSM to synthesize novel views for missing or occluded regions based on the generated views. Given input views, geometry maps, and camera poses, the model synthesizes new viewpoints from arbitrary target poses, effectively densifying the texture observations. In contrast to UV-based methods (Yu et al., 2023; Cheng et al., 2024; Yu et al., 2024; Zeng et al., 2024a), which suffer from discontinuities and topological ambiguities in UV space, our approach performs view-space completion, enabling seamless and globally consistent texture generation.

The contributions of this work are summarized as follows:

- We propose an end-to-end multi-branch framework for high-quality PBR texture generation, where a multi-view illumination context branch captures consistent lighting priors to alleviate data imbalance and reliance on imperfect intermediates in two-stage designs.
- We introduce a lighting-aware material attention mechanism that conditions albedo and metallic–roughness generation on shared illumination priors, enabling disentangled material reasoning and improving physical plausibility.
- We propose an advanced texture inpainting strategy, leveraging LVSM to extend the generated views to a denser set for seamless and globally consistent texture completion.
- Extensive experiments demonstrate that our method surpasses existing state-of-the-art open-source and commercial methods.

## 2 RELATED WORK

**Texture Generation.** The advances of foundation models have opened new directions for automating texture generation in 3D content creation. Early works leverage 2D diffusion priors via Score Distillation Sampling (SDS) to optimize 3D assets (Poole et al., 2022; Lin et al., 2023; Wang et al., 2023; Po & Wetzstein, 2024; Metzer et al., 2022; Chen et al., 2023a; Khalid et al., 2022; Michel et al., 2021; Chen et al., 2022). These approaches iteratively optimize renderings of 3D shapes to align with the distribution learned by pre-trained diffusion models (Podell et al., 2023; Rombach et al., 2021; Esser et al., 2024; Lin et al., 2024), but the results are often over-saturated and unregulated for 3D shapes, making them inapplicable for practical use. To enhance geometric fidelity, some methods (Yu et al., 2023; Bensadoun et al., 2024; Cao et al., 2023; Liu et al., 2024a) incorporate explicit 3D features, such as vertex positions, normals, or depths, to progressively inpaint the mesh across pre-defined views. Although the geometry fidelity is improved, the results are deteriorated by the synchronization process of multi-view latents. Another line of research (Yu et al., 2023; Zeng et al., 2024a; Bensadoun et al., 2024; Yu et al., 2024) projects the 3D point cloud information and supervises the training in the UV space, addressing the occlusion problem in the multi-view approaches. However, they introduce topological ambiguity inherent from the UV representation, deteriorating the capability of the diffusion model to generate high-fidelity textures.

More recently, methods like MV-Adapter (Huang et al., 2024c) and Hunyuan3D-Paint (Zhao et al., 2025) have shown promising results in generating globally consistent textures via multi-view attention (Li et al., 2024; Kant et al., 2024; Huang et al., 2024d; Shi et al., 2023; Wang & Shi, 2023; Feng et al., 2025). These methods efficiently leverage the capability of pre-trained diffusion models and the 3D geometry condition, ensuring both realistic results and spatial consistency.

**PBR Texture Generation.** Recent approaches leverage pre-trained diffusion models to generate PBR materials for 3D assets. Early works employ SDS for PBR generation and typically (Chen et al., 2023c; Zhang et al., 2024c; Liu et al., 2024b; Youwang et al., 2024a; Wu et al., 2023; Xu et al., 2023;

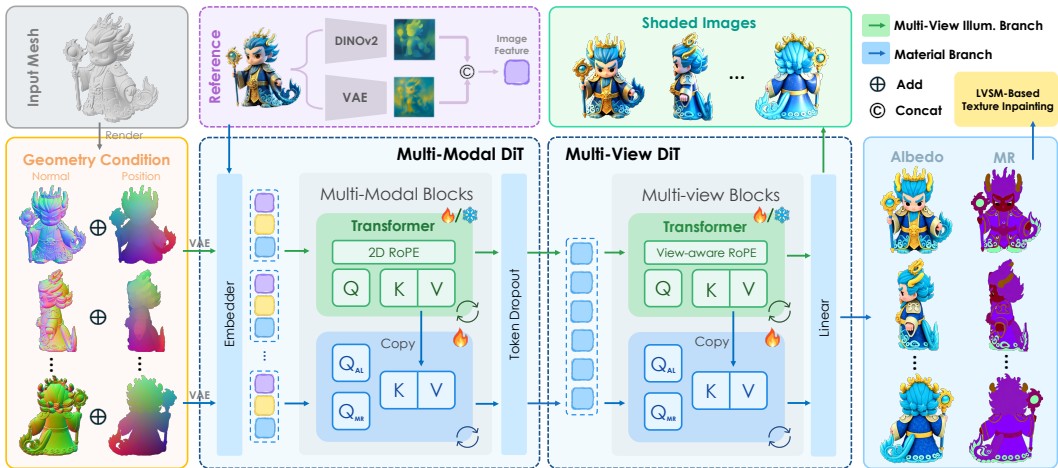

Figure 3. **Overview of LumiTex.** We first train a multi-view illumination-consistent base model to generate shaded images. Then, we freeze this branch and utilize its shaded features to train material branches for high-fidelity PBR texture generation. Finally, our geometry-guided LVSM synthesizes novel views from novel perspectives to enable seamless, view-consistent texture inpainting.

Yeh et al., 2024) incorporate the BRDF in the diffusion process to learn material properties. Methods like Material-Anything (Huang et al., 2024a) and CLAY (Zhang et al., 2024a) iteratively denoise and synchronize latents across multiple viewpoints. TexGaussian (Xiong et al., 2024) leverages octant-aligned 3DGS (Kerbl et al., 2023) for real-time PBR texture synthesis. Other approaches (Zhang et al., 2024b; Dang et al., 2024; Wang et al., 2024; Fang et al., 2024) leverage LLMs to improve semantic alignment with retrieval-augmented generation. However, these methods struggle with physically grounded decomposition, view consistency, or accurate separation of albedo and MR. Some approaches (He et al., 2025; Zhu et al., 2024; Vainer et al., 2024; Vecchio et al., 2024; Sartor & Peers, 2023) fine-tune image models to generate multi-channel PBR materials, but the domain gap between albedo and MR is not well addressed.

**Image Intrinsic Decomposition.** Material decomposition aims to estimate intrinsic materials from image inputs with unknown lighting, serving as a fundamental task in material understanding. Early works (Wimbauer et al., 2022; Sang & Chandraker, 2020; Boss et al., 2020; Yi et al., 2023) formulate this task as an optimization problem, recovering the PBR materials from the images by solving the rendering equation (Kajiya, 1986). Recently, generative approaches (Chen et al., 2024; Hong et al., 2024; Kocsis et al., 2024; Li et al., 2025b; Zhu et al., 2022; Huang et al., 2024b) utilize diffusion models to decompose materials from images with promising results. RGB-X (Zeng et al., 2024b) and DiffusionRenderer (Liang et al., 2025a) propose a unified forward and inverse process for image synthesis and material decomposition. However, estimating individual properties from a single image with unknown illumination is ill-posed due to the inherent ambiguity between illumination and materials. Our work designs a multi-view pipeline that incorporates illumination context, enabling physically plausible and coherent material generation.

## 3 METHOD

Given an input mesh with a reference image $I$, our goal is to generate $N$ view-consistent PBR images and achieve seamless material textures. We describe the overall pipeline architecture in Section 3.1. As shown in Fig. 3, we first train a multi-view illumination context branch that reconstructs multi-view shaded images to capture consistent lighting cues as an explicit illumination context. This context then guides the albedo and MR branches through a lighting-aware material attention mechanism to generate multi-view material maps (Section 3.2). Finally, a geometry-guided LVSM is trained to synthesize $M$ novel views from generated $N$ views for texture inpainting (Section 3.3).

## 3.1 MULTI-VIEW PBR GENERATION TRANSFORMER

Our model integrates a multi-modal Transformer (MM-T) and a multi-view Transformer (MV-T). The multi-modal transformer fuses diverse modalities for each view, while the multi-view transformer enforces consistency across views. The detailed operations are presented below.

**Multi-Modal DiT.** The MM-T is designed to integrate geometry information, reference appearance, and material semantics (albedo or MR) for each view. Specifically, for each view $i = 1, ..., N$, we concatenate tokens derived from the input image encoded by VAE and DINOv2 (Oquab et al., 2023), the mesh geometry (normal $N_i$ and canonical coordinate $C_i$) encoded by VAE, and learnable material embeddings $e$, as well as view latent $z_i$, and feed them into a stack of $l_1$ transformer blocks:

$$\mathbf{T}_{\text{geo}} = \text{Linear}_{\text{geo}}\big[\text{VAE}(N_i) \oplus \text{VAE}(C_i)\big] \in \mathbb{R}^{N \times L \times C}, \tag{1}$$

$$\mathbf{T}_{\text{img}} = \text{Linear}_{\text{img}}\big[\text{VAE}(I); \text{DINO}(I)\big] \in \mathbb{R}^{1 \times 2L \times C}, \tag{2}$$

$$z_i^{l_1} = \text{MM-T}^{l_1}\big(z_i, \mathbf{T}_{\text{img}}, \mathbf{T}_{\text{geo}}, e\big) \in \mathbb{R}^{N \times 3L \times C}. \tag{3}$$

**Multi-View DiT.** After fusing multi-modal features in the first stage, we discard the image and domain tokens, and shift focus to enforcing cross-view consistency in the second stage. Specifically, we concatenate their latent tokens from $N$ views into a unified sequence that is processed by a sequence of $l_2$ transformer blocks, allowing each token to interact with others for global consistency:

$$\{\hat{z}_i\}_{i=1}^N = \text{MV-T}^{l_2}(z_1^{l_1}, z_2^{l_1}, ..., z_N^{l_1}) \in \mathbb{R}^{NL \times C}, \tag{4}$$

where $\hat{z}_i$ is the denoised latent for the $i$-th view, $L$ is the token length, and $C$ is the feature dimension. The whole model is trained end-to-end using a flow matching loss on multi-view images.

## 3.2 LIGHTING-AWARE MATERIAL ATTENTION

Multi-view shaded images provide rich illumination cues for high-fidelity PBR reconstruction, serving as a reliable prior for high-fidelity PBR reconstruction. Following this insight, we design a multi-branch generative framework, where a multi-view illumination context branch provides shaded embeddings for material reasoning. The lighting-aware material branch then consumes this context to produce physically plausible PBR maps. This framework alleviates the data scarcity problem in PBR texture generation, mitigates the domain gap between albedo and MR maps, and yields more physically consistent material predictions.

**Multi-View Illumination Context Branch.** Unlike prior two-stage works that rely on multi-view shaded images as intermediates, we introduce a multi-view illumination context branch to learn the shaded embeddings, as illustrated in Fig. 3. Specifically, this branch is trained to reconstruct multi-view shaded images to ensure the learned embeddings capture consistent illumination across views. To model view-dependent illumination effects, shaded latent tokens $S = \{s_i\}_{i=1}^{NL}$ from all views are encoded with a view-aware RoPE (Su et al., 2024), which preserves both spatial alignment and view identity. Then, we perform cross-view attention to produce shaded keys and values that condition the material branches:

$$s_i = \sum_j \text{Softmax}_j \big(q_i k_j^T + \phi(t, i, j)\big) v_j, \quad K_{\text{shaded}} = W_K S, \quad V_{\text{shaded}} = W_V S, \tag{5}$$

where $\phi(t, i, j)$ is a view-specific positional embedding tied to the index of query view $t$, encoding illumination relationships between views. We provide the details in Sec. A.2 of the Appendix.

This design alleviates the data imbalance issue in open-source 3D datasets, where high-quality PBR maps are scarce. In this way, even data that lack plausible MR maps can be utilized, as they still provide supervision for multi-view illumination reconstruction under physically consistent materials.

**Lighting-Aware Material Branch.** Some works generate PBR materials via jointly modeling the albedo and MR maps, overlooking the domain gap between these two modalities. To address this problem, we leverage the illumination context learned from the previous branch and guide albedo and MR generation separately, while conditioning both on the same shaded features to maintain the consistency of the two branches. Specifically, we introduce a *lighting-aware material attention* mechanism in which albedo and MR branches perform shaded-guided cross-attention. The shaded

keys $\boldsymbol{K}_{\text{shaded}}$ and values $\boldsymbol{V}_{\text{shaded}}$ supply illumination cues to inform material-specific queries:

$$\text{Attn}_{\text{albedo}} = \text{Softmax}\left(\frac{\boldsymbol{Q}_{\text{albedo}}\boldsymbol{K}_{\text{shaded}}^T}{\sqrt{d}}\right) \cdot \boldsymbol{V}_{\text{shaded}}, \tag{6}$$

$$\text{Attn}_{\text{mr}} = \text{Softmax}\left(\frac{\boldsymbol{Q}_{\text{mr}}\boldsymbol{K}_{\text{shaded}}^T}{\sqrt{d}}\right) \cdot \boldsymbol{V}_{\text{shaded}}. \tag{7}$$

This design provides a rich illumination context for the decoding process, enabling the model to separate reflectance properties from lighting effects and avoid the error accumulation problem in the two-stage design. By complementarily attending to shaded priors, the albedo branch emphasizes diffuse consistency while the MR branch captures specular characteristics, together improving both the physical plausibility and multi-view coherence of generated PBR textures.

### 3.3 TEXTURE INPAINTING AS NOVEL VIEW SYNTHESIS

Texture inpainting is a post-processing step that fills missing or occluded regions to ensure seamless integration with surrounding textures. Previous methods usually perform inpainting on partially textured meshes in the UV space. However, due to inherent UV discontinuities and topological ambiguities, the inpainted regions often fail to align with their surrounding areas, particularly when the UV mapping is highly fragmented, as is common for shapes produced by generative models.

Incomplete textures typically result from limited surface coverage in the generated sparse multi-view images. To address this, we aim to synthesize a dense set of views that fully cover the object's surface, enabling seamless texture completion through direct back-projection. To this end, we train a geometry-guided variant of LVSM (Jin et al., 2025), a scalable large view synthesis model known for high-quality novel view generation, to infer additional views that cover previously unobserved regions of the mesh. These synthesized views, combined with the initial ones, form a dense set that is projected back to UV space, resulting in a seamless and complete texture. The proposed framework is illustrated in Fig. 4.

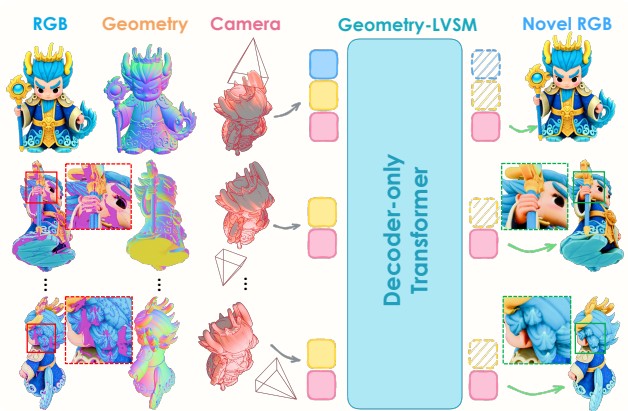

Figure 4. **Texture Inpainting with Geometry-guided LVSM (Jin et al., 2025).** Our model infers dense novel views from sparse inputs and aligns with the geometry.

**Novel View Generation.** Given $N$ generated sparse view images $\{\boldsymbol{I}_i\}_{i=1}^N$, pixel-aligned Plücker ray maps $\{\boldsymbol{P}_i\}_{i=1}^N$ that encode camera intrinsics and extrinsics, and geometry conditions $\{\boldsymbol{G}_i\}_{i=1}^N$, we infer additional $M$ target views with conditions $\{\boldsymbol{P}_i^t\}_{i=1}^M$ and $\{\boldsymbol{G}_i^t\}_{i=1}^M$ to inpaint the occluded regions. To effectively embed conditions, we tokenize and map the inputs into a unified representation with a linear layer. Formally,

$$\boldsymbol{x}_i = \text{MLP}([\boldsymbol{P}_i, \boldsymbol{G}_i, \boldsymbol{I}_i]), \quad \boldsymbol{x}_i^t = \text{MLP}([\boldsymbol{P}_i^t, \boldsymbol{G}_i^t]) \in \mathbb{R}^d, \tag{8}$$

where $d$ is the feature dimension, $\boldsymbol{x}_i$ represents the set of condition tokens, and $\boldsymbol{x}_i^t$ denotes the set of target tokens. Following LVSM (Jin et al., 2025), we employ a decoder-only transformer to infer target views from input tokens that avoid explicit 3D representation to minimize the inductive bias:

$$\{\boldsymbol{y}_i\}_{i=1}^N, \{\boldsymbol{y}_i^t\}_{i=1}^M = \text{Transformer}\left(\{\boldsymbol{x}_i\}_{i=1}^N, \{\boldsymbol{x}_i^t\}_{i=1}^M\right), \tag{9}$$

where the output condition tokens $\boldsymbol{y}_i$ and target tokens $\boldsymbol{y}_i^t$ are updated from the corresponding inputs. Then, we discard condition tokens and map target tokens to the RGB space with an MLP, followed by reshaping to form the final predicted images $\hat{\boldsymbol{I}}^t$:

$$\hat{\boldsymbol{I}}_1^t, \ldots, \hat{\boldsymbol{I}}_M^t = \text{Reshape}\left(\text{MLP}(\boldsymbol{y}_1^t, \ldots, \boldsymbol{y}_M^t)\right). \tag{10}$$

To further improve the quality of the synthesized novel views, we adopt the test-time training method proposed in Zhang et al. (2025). The geometry-guided LVSM generates two dense sets of albedo

and MR separately for texture inpainting, where MR is encoded in the ORM convention with the occlusion channel set to a constant value.

**View Selection.** Inspired by Zhao et al. (2025), we select target views from a predefined dense set $\mathcal{V} = \{v_1, ..., v_K\}$. We first project the generated $N$ views into the UV space to obtain an incomplete texture. Then, we greedily rank the views in $\mathcal{V}$ by the area of uncovered UV regions they observe and select the top $M$ as target views.

## 4 EXPERIMENTS

In this section, we evaluate our method with both open-source and commercial state-of-the-art methods, including shaded texture generation, PBR generation, and texture inpainting methods. We conduct comprehensive qualitative and quantitative comparisons, and a user study with 3D modelers, to demonstrate that our generated PBR textures align well with human perceptual preferences.

### 4.1 IMPLEMENTATION DETAILS

We curate a dataset from Objaverse and Objaverse-XL (Deitke et al., 2023b;a), containing 92K objects as our training set. For each 3D object, we sample cameras from 30 views and render the object with 3 environment maps. We also render multi-view albedo, metallic, roughness maps, and HDR images at a resolution of $1024 \times 1024$. For the model training, we initialize our DiT from FLUX.1-dev (BlackForestLabs, 2024), utilizing the flow matching as the training objective. We first train our shaded model on $512 \times 512$ and then scale up to $768 \times 768$. The entire training procedure requires approximately 106 GPU days. We provide datasets and training details in Sec. A.2.

### 4.2 COMPARISON WITH EXISTING METHODS

**Baselines.** We compare our method with comprehensive texture synthesis baselines to demonstrate its effectiveness. The baselines include shaded texture synthesis: SyncMVD (Liu et al., 2024a), MV-Adapter (Huang et al., 2024c), Step1X-3D (Li et al., 2025a), UniTEX (Liang et al., 2025b); and PBR texture synthesis: Paint-it (Youwang et al., 2024b), DreamMat (Zhang et al., 2024c), and Hunyuan3D-2.1 (Team, 2025a) (MaterialMVP (Huang et al., 2025) enhanced with purchased dataset). Additionally, we compare LumiTex with proprietary methods such as Meshy-5 (Meshy, 2025) and Tripo AI v2.5 (TripoAI, 2025). We improve the original text-conditioned SyncMVD (Liu et al., 2024a) by incorporating the SDXL-base model (Podell et al., 2023) (the vital component) and an IP-Adapter (Ye et al., 2023) to align with an image-to-texture task and compare it (referred to as SyncMVD-IPA) with our approach. We generate some image prompts using GPT-4o and obtain meshes from Hunyuan3D-2.5 (Team, 2025b) and Objaverse (Deitke et al., 2023b). For the texture inpainting, we compare our method with Paint3D (Zeng et al., 2024a) and TexGen (Yu et al., 2024).

**Evaluation Metrics.** We follow Hunyuan3D (Team, 2025a;b; Zhao et al., 2025) to use FID (Heusel et al., 2017), CLIP-FID, and LPIPS (Zhang et al., 2018) to evaluate texture fidelity. CLIP Maximum-Mean Discrepancy (CMMD) (Jayasumana et al., 2024) assesses the diversity of the generated texture details, and CLIP-I (Radford et al., 2021) measures prompt alignment. We further assess relighting quality under novel lighting as an indicator of the physical accuracy of the generated PBR materials.

### 4.3 QUALITATIVE RESULTS

We qualitatively compare LumiTex with recent baselines in texture generation and inpainting. LumiTex surpasses baselines in physical realism, lighting decoupling, and prompt fidelity. In terms of realism, LumiTex exhibits realistic appearances under novel lighting, while others often copy input highlights as in Fig. 5, failing to produce realistic reflections due to the absence of PBR materials. As shown in Figs. 5 and 6, LumiTex effectively decouples lighting effects to ensure that models interact correctly with environment-dependent illumination. In contrast, baselines often produce diffuse maps with baked-in lighting, which deteriorates realism and hinders 3D asset reuse. Compared to SyncMVD, Step1X-3D, and DreamMat, which exhibit color shifts, our method produces semantically faithful PBR textures that preserve fine details. LumiTex can also generate high-quality textures across complex shapes and diverse object categories (see Fig. 1), retaining realistic appearances under diverse lighting (see Fig. 8), supporting downstream applications like IP design.

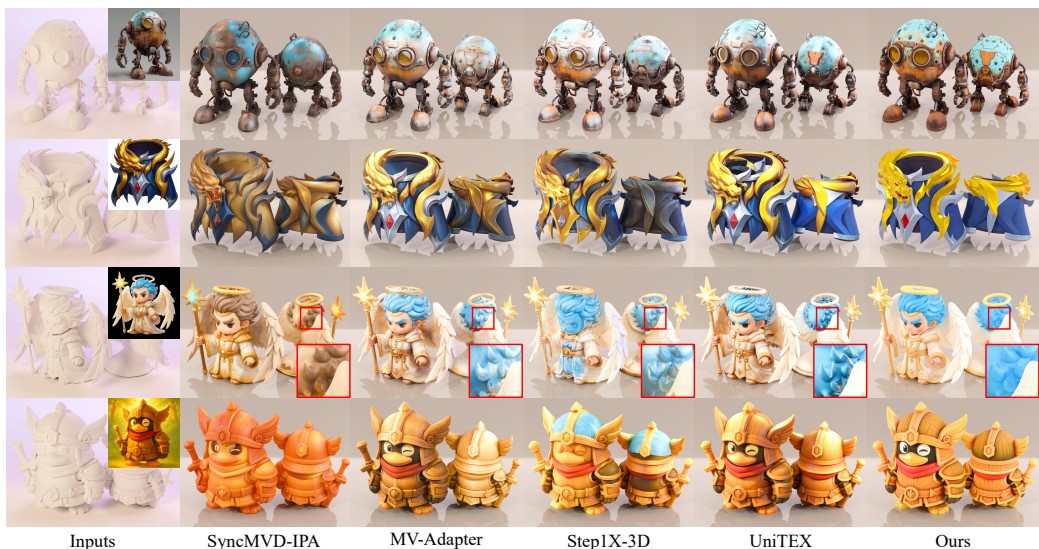

Figure 5. **Qualitative Results on Texture Generation Methods**. Our method generates plausible materials for relighting, avoids baked-in lighting, and is robust under diverse reference illuminations.

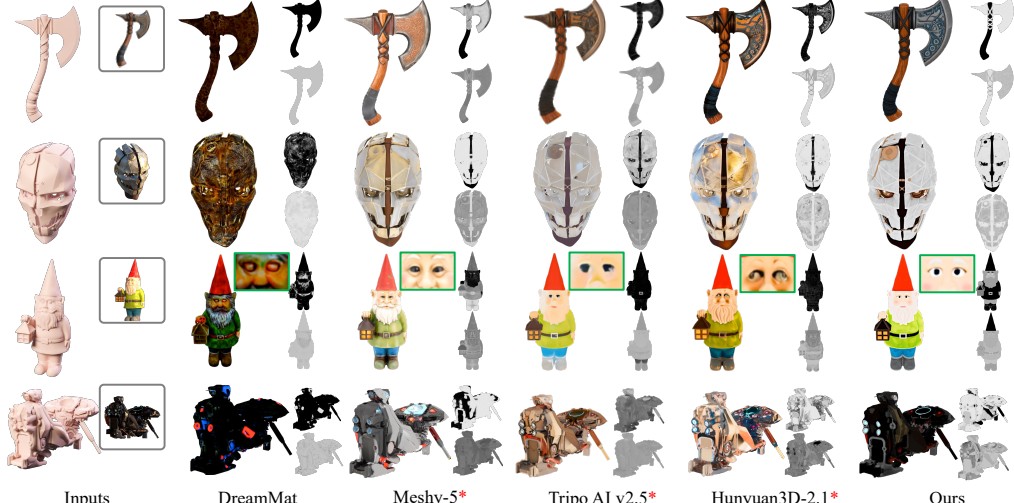

Figure 6. **Qualitative Results on PBR Generation Methods**. Our method generates high-fidelity PBR materials, avoids light baking, and achieves competitive PBR maps compared to state-of-the-art open-source and commercial methods. Each object has: the albedo on the left, the metallic on the top right, and the roughness on the bottom right. * denotes the method trained on private datasets.

In Fig. 7, we compare texture inpainting results across baselines. Paint3D completes textures with a UV refinement model but often produces over-smoothed and semantically inconsistent results, as spatially unrelated regions are treated equally in the UV domain. For example, the Beetle's underside and wheels are inpainted without spatial awareness, leading to texture bleeding. TexGen mitigates this issue via 3D spatial encoding, yet still suffers from disrupted semantic coherence and seams, due to the inherent topological ambiguities of UV mapping. Our method, conditioned on 2D views and 3D cameras, avoids the adverse effects caused by UV representation. By unifying condition and target views in a framework, our method preserves fine details while ensuring global consistency.

## 4.4 QUANTITATIVE RESULTS

We evaluate 133 objects excluded from training on two aspects: texture quality and relighting evaluation. Texture quality assesses fidelity and alignment with the reference image, while relighting

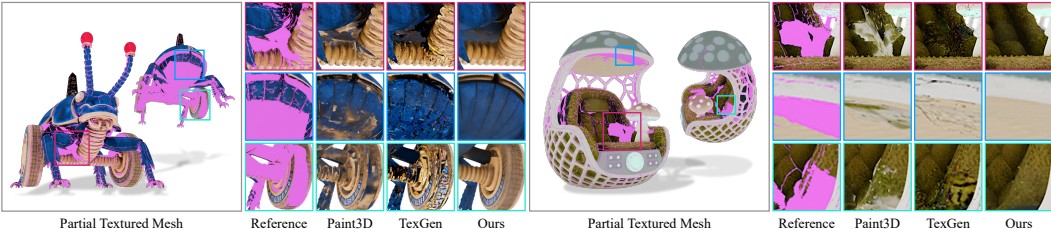

Figure 7. **Comparison with Texture Inpainting Methods.** Our approach effectively recovers local details and exhibits greater robustness than other methods, producing semantically coherent results without visible seams.

Table 1. Quantitative comparison with state-of-the-art methods. We compare two classes of methods, texture-only generation and PBR texture generation. Our method achieves the best performance compared with both classes. * denotes the method trained on private datasets.

| Method | Type | Texture Evaluation | | | | | Relighting Evaluation | | | | |
| --- | --- | --- | --- | --- | --- | --- | --- | --- | --- | --- | --- |
| | | FID↓ | CLIP-FID↓ | CMMD↓ | CLIP-I↑ | LPIPS↓ | FID↓ | CLIP-FID↓ | CMMD↓ | CLIP-I↑ | LPIPS↓ |
| SyncMVD-IPA (Liu et al., 2024a) | Texture | 222.1 | 21.10 | 1.8263 | 0.9187 | 0.2504 | 149.1 | 18.04 | 0.7394 | 0.9101 | 0.1202 |
| MV-Adapter (Huang et al., 2024c) | Texture | 237.3 | 24.95 | 2.4510 | 0.9022 | 0.2574 | 123.2 | 13.82 | 0.5405 | 0.9246 | 0.1034 |
| Step1X-3D (Li et al., 2025a) | Texture | 240.9 | 24.32 | 2.2090 | 0.9053 | 0.2540 | 120.0 | 12.90 | 0.4038 | 0.9288 | 0.1000 |
| UniTEX (Liang et al., 2025b) | Texture | 230.7 | 22.20 | 1.9891 | 0.9133 | 0.2473 | 124.8 | 13.50 | 0.4707 | 0.9282 | 0.0974 |
| Paint-it (Youwang et al., 2024b) | PBR | 293.3 | 35.50 | 3.4137 | 0.8648 | 0.3769 | 162.9 | 26.77 | 1.2514 | 0.8666 | 0.1564 |
| DreamMat (Zhang et al., 2024c) | PBR | 231.6 | 25.49 | 2.1722 | 0.9016 | 0.2816 | 160.1 | 19.97 | 0.8386 | 0.8983 | 0.1346 |
| Hunyuan3D-2.1*(Team, 2025a) | PBR | 196.6 | 18.84 | 1.7195 | 0.9268 | 0.2413 | 103.7 | 10.89 | 0.3610 | 0.9420 | **0.0808** |
| Ours | PBR | **160.8** | **14.89** | **1.3669** | **0.9417** | **0.1903** | **99.6** | 10.63 | **0.3151** | **0.9436** | 0.0831 |

evaluation involves rendering each object from 32 views sampled on the Fibonacci sphere with random environment maps and comparing them against ground-truth renderings. As shown in Tab. 1, our method surpasses both texture-only and PBR-based baselines, achieving better FID and LPIPS for visual fidelity, lower CMMD for richer and more diverse texture generation, and stronger semantic alignment with the reference images as reflected by higher CLIP-I and lower CLIP-FID scores.

To evaluate perceptual quality, we conduct a user study, where 23 3D modelers rate results (ranging from 1 to 5) generated from different methods on five criteria that are difficult to quantify: rendering quality, albedo, roughness, metallic accuracy, and texture completeness, with input images and meshes provided for ref-

Table 2. User study results on rendering quality, completeness, and PBR material accuracy.

| Method | Quality↑ | Compl.↑ | Diffuse↑ | Metallic↑ | Rough.↑ |
| --- | --- | --- | --- | --- | --- |
| SyncMVD (Liu et al., 2024a) | 2.29 | 3.27 | 2.39 | – | – |
| MV-Adapter (Huang et al., 2024c) | 2.65 | 3.06 | 2.65 | – | – |
| Step1X-3D (Li et al., 2025a) | 2.70 | 2.98 | 2.58 | – | – |
| UniTEX (Liang et al., 2025b) | 2.96 | 2.98 | 2.75 | – | – |
| Paint-it (Youwang et al., 2024b) | 2.27 | 2.97 | 2.19 | 2.46 | 2.57 |
| DreamMat (Zhang et al., 2024c) | 2.05 | 2.92 | 2.58 | 2.40 | 2.33 |
| Hunyuan3D-2.1*(Team, 2025a) | 3.69 | 3.98 | 3.57 | 3.34 | 3.61 |
| Ours | **4.48** | **4.61** | **4.34** | **4.14** | **4.07** |

erence. As shown in Tab. 2, our method outperforms all baselines across all criteria, demonstrating strong alignment with human preference.

## 4.5 ABLATION STUDY

**One-Stage Generation.** To validate the effectiveness of our end-to-end pipeline, we compare it to a two-stage variant that first generates multi-view shaded images, followed by a PBR decomposition model. This variant is fine-tuned from our illumination context branch and IDArb (Li et al., 2025b). As shown in Fig. 9(a), it often produces inaccurate material predictions, such as excessive metallicity, plastic-like surfaces, or overly uniform albedos, likely due to error accumulation across stages. In contrast, our unified design produces more realistic results, showing stronger fidelity for this task. **Multi-Branch Generation.** To evaluate our disentangled multi-branch design, we compare it with a multi-channel variant that jointly predicts albedo and MR in a unified output space, following prior works (He et al., 2025; Zhang et al., 2024a). As shown in Fig. 9(b), this joint prediction often results in inaccurate outputs, especially in metallic regions. Our illumination-guided, material-specific design maintains channel separation and produces more physically and semantically accurate results. **Multi-View Illumination Context Branch.** We show the importance of illumination context branch by training a material-only variant on the same dataset until convergence. As shown in Fig. 9(c), the variant fails to produce accurate MR maps in metallic regions, resulting in a plastic-like appearance under relighting. The illumination context contributes greatly to accurate PBR generation.

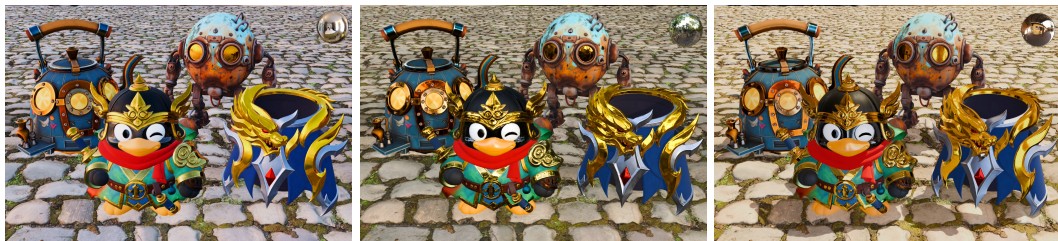

Figure 8. **Relighting Results.** We generate 3D assets from various input images and relight them under diverse environments to demonstrate the physical plausibility of our PBR textures.

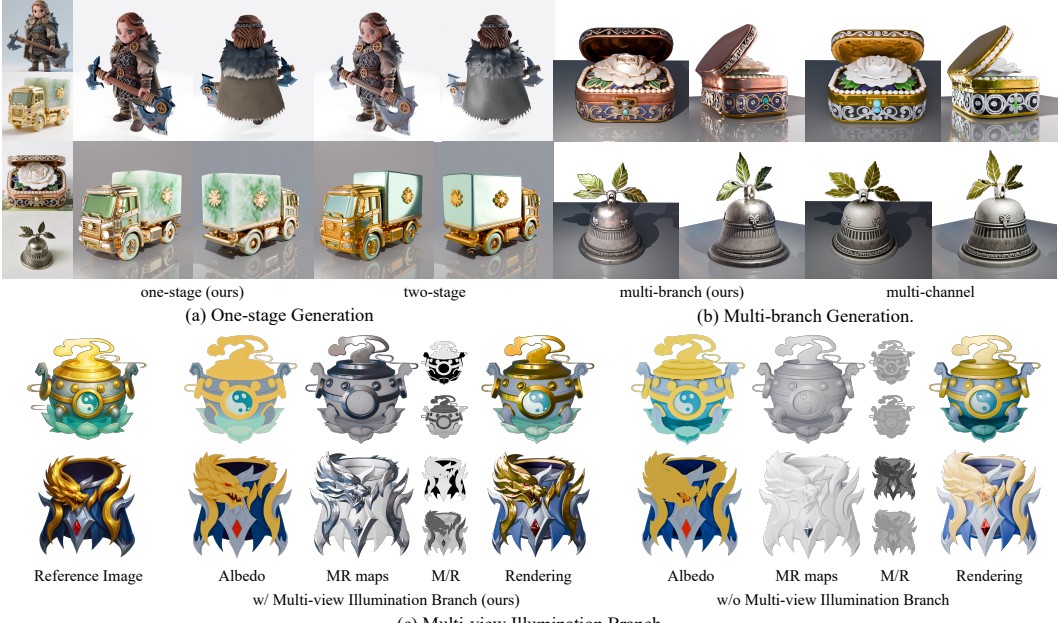

Figure 9. We ablate the one-stage generation, multi-branch generation, and multi-view illumination context branch (top: metallic, down: roughness) to validate the effectiveness of our method.

## 5 CONCLUSION

We present LumiTex, an end-to-end multi-branch pipeline for high-fidelity PBR texture generation. By combining a multi-view illumination context branch with a novel lighting-aware material attention mechanism, LumiTex enables physically plausible PBR map generation. To ensure global surface coverage and coherence, we train a geometry-guided LVSM for texture inpainting. Extensive experiments demonstrate that LumiTex outperforms existing methods in texture quality, semantic alignment, and relighting fidelity, offering a practical solution for PBR texture generation.

## 6 ACKNOWLEDGMENT

We thank the Hunyuan3D team for their inspiring works (Team, 2025a;b; He et al., 2025; Feng et al., 2025). The visual design of our teaser is built upon creative assets from Hunyuan3D 2.5. For the experimental comparisons, the 3D assets are sourced from the Hunyuan3D platform to ensure a direct and consistent evaluation. The work was supported in part by Guangdong Provincial Outstanding Youth Fund with No. 2023B1515020055, the Shenzhen Outstanding Talents Training Fund 202002, the NSFC with Grant No. 62293482, the Guangdong Research Projects No. 2017ZT07X152 and No. 2019CX01X104, the Guangdong Provincial Key Laboratory of Future Networks of Intelligence (Grant No. 2022B1212010001), and the Shenzhen Key Laboratory of Big Data and Artificial Intelligence (Grant No. SYSPG20241211173853027), the Guangdong Province Radio Science Data Center.

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

## A APPENDIX

### A.1 MORE VISUAL RESULTS OF LUMITEX

**Multi-View PBR Material Results.** To evaluate the generalization capability of LumiTex in estimating multi-view G-buffers from in-the-wild images, we present results based on real-captured images (Fig. 10) and AI-generated images (Fig. 11). As illustrated, LumiTex produces consistent multi-view RGB renderings and corresponding materials in both cases, demonstrating strong robustness to images beyond the training distribution. The material renderings are scaled to avoid pure white regions, which may visually blend with the background and hinder clarity.

**Results on Real-World Scanned Meshes.** We further evaluate LumiTex on real-captured scenes with casual image prompts and noisy geometry reconstructed from photogrammetry. As shown in Fig. 13, LumiTex generates high-quality PBR textures that remain physically plausible and consistent across novel views, despite challenging lighting conditions and imperfect inputs. Notably, our model preserves fine details and produces realistic material, highlighting its robustness in real-world scenarios without requiring clean geometry or controlled lighting.

**Relighting Comparisons.** To demonstrate the quality of our PBR textures, we compare relighting results against several state-of-the-art PBR generation methods. As shown in Fig. 12, LumiTex consistently generates realistic, lighting-consistent renderings across diverse scenes and materials, and robustly handles structured patterns like text. In contrast, competing methods often exhibit inaccurate material predictions or texture artifacts under novel lighting. Additionally, we provide supplementary videos demonstrating dynamic relighting, including static objects under rotating environment lighting and rotating objects under fixed illumination. These visualizations showcase the photorealism and physical plausibility of our results.

**More PBR Decomposition Results.** We report the G-buffer results of the teaser in Fig. 15, containing 38 assets with both real and AI generated meshes and references.

**Robustness under Extreme Conditions.** To further assess generalization, we evaluate LumiTex on inputs exhibiting highly reflective materials and strong backlighting, two challenging scenarios where shading leakage and highlight imprinting are common. As shown in Fig. 16, our method consistently maintains clean albedo maps and stable MR predictions without baked-in reflections, while baseline methods struggle under the same conditions.

### A.2 IMPLEMENTATION DETAILS

In this section, we elaborate on the implementation details, including the dataset construction and augmentation, the details of the implementation of the multi-view PBR generation transformer, and the geometry-conditioned LVSM for texture inpainting.

**Dataset Details.** We curate a dataset from Objaverse and Objaverse-XL (Deitke et al., 2023b;a), comprising 92K 3D objects for training both our PBR generation model and the large view synthesis model. For each object, we sample 6 predefined camera viewpoints with the following elevation–azimuth pairs for PBR generation transformer training: $\{20°, 0°\}$, $\{20°, 90°\}$, $\{20°, 180°\}$, $\{20°, 270°\}$, $\{90°, 0°\}$, and $\{-90°, 0°\}$. Then, we sample 24 random camera viewpoints to train our LVSM-based texture inpainting model. To ensure dense and diverse coverage, each selected view is required to differ from all others by at least $5°$ in either azimuth or elevation. Each object is rendered under 3 randomly selected environment maps from a lighting database to provide diverse illumination. We use Blender Cycles and Nvdiffrast with custom shaders to generate multi-view shaded images and G-buffers (including normal, XYZ, albedo, metallic, and roughness maps). All images are rendered at a resolution of $1024 \times 1024$ and downsampled to $512 \times 512$ and $768 \times 768$ for multi-scale training.

**Data Augmentation.** To enhance the robustness of our method for real-world applications, such as user-generated content (UGC) platforms where inputs may be AI-generated or casually captured, we employ several commonly used data augmentation strategies. To simulate the variability of user-provided reference images, which often deviate from the canonical viewpoints in our training set, we apply random transformations including scaling, rotation, translation, perspective warping, and lumination scaling with probabilities 0.1, 0.25, 0.25, 0.1, 0.1, respectively.

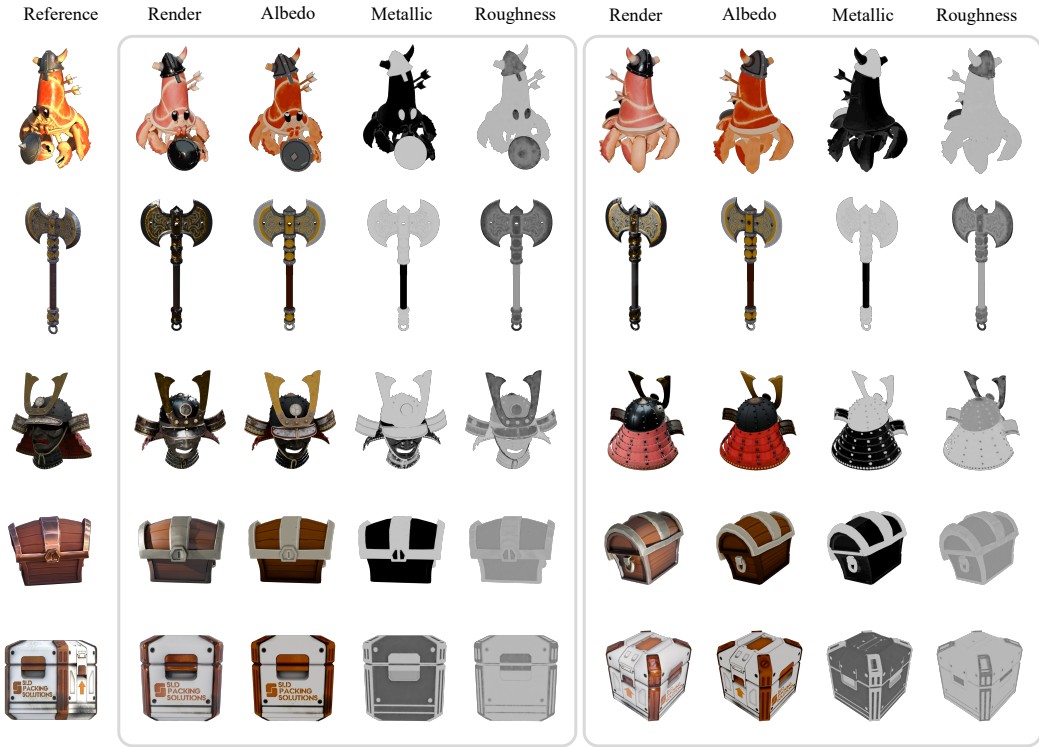

Figure 10. **Multi-view rendered images and materials generated by LumiTex.** All of the input images are real captured images. We present novel view renderings, along with corresponding albedo, metallic, and roughness maps, from two different views.

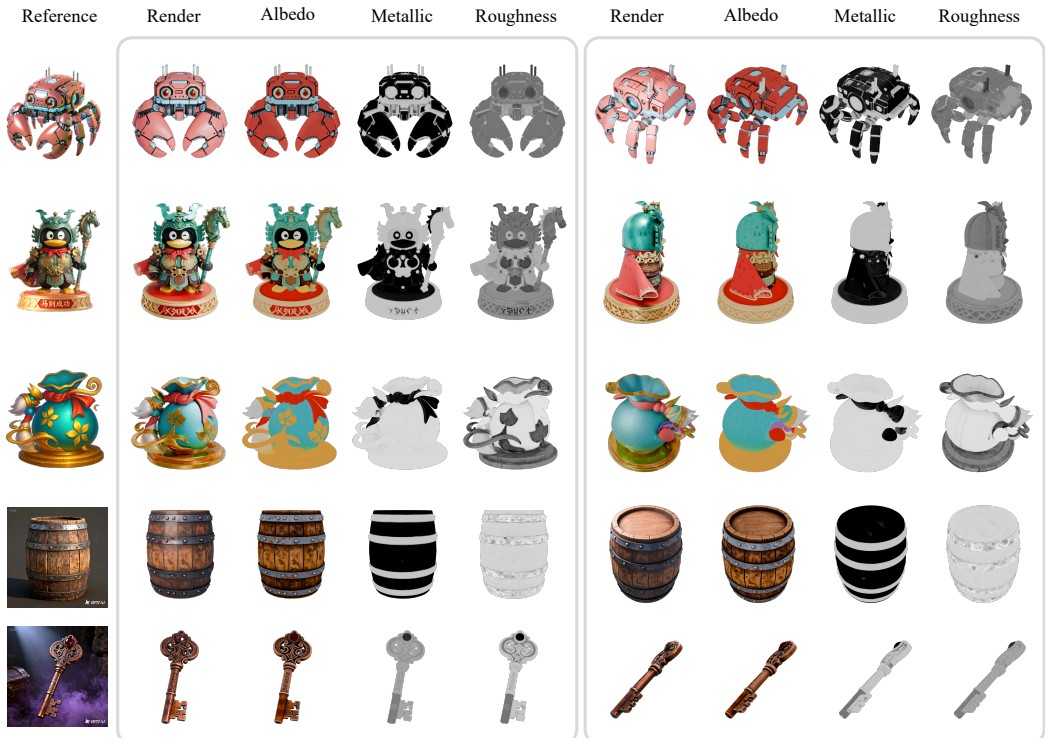

Figure 11. **Multi-view rendered images and materials generated by LumiTex.** All of the input images are real AI-generated images. We present novel view renderings, along with corresponding albedo, metallic, and roughness maps, from two different views.

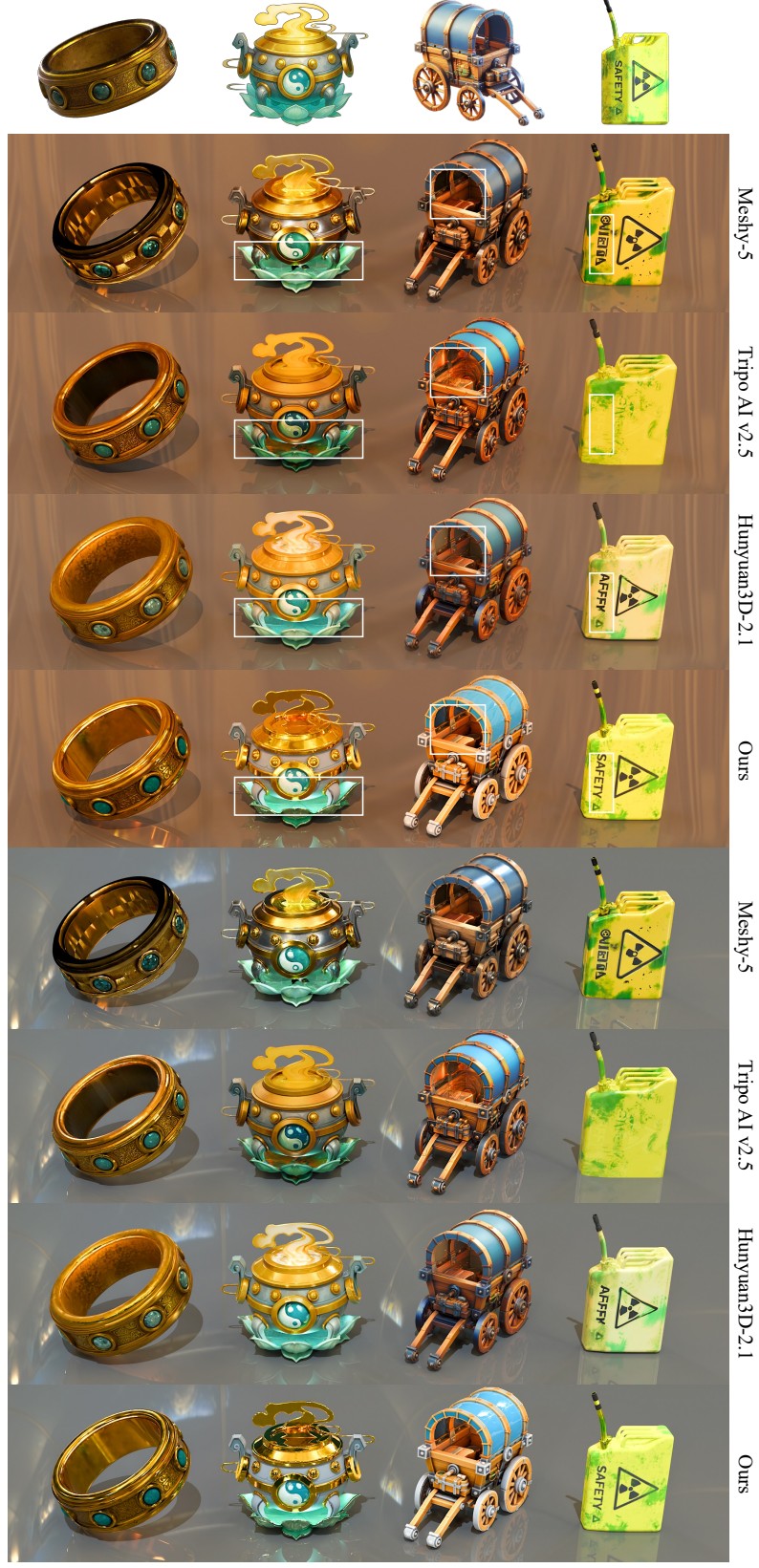

Figure 12. **Relighting Comparisons.** Relighting results of state-of-the-art PBR generation methods under two novel lighting conditions. The top row shows the reference images.

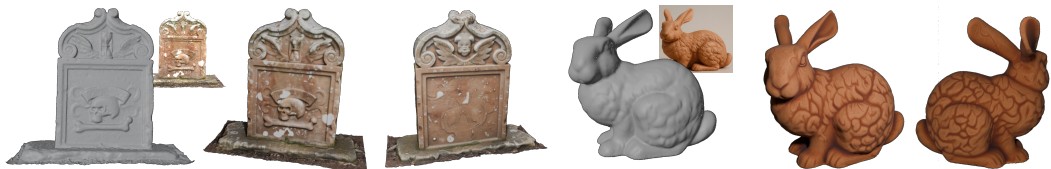

Figure 13. **Generalization to Real-world Scanned Meshes.** LumiTex generates consistent PBR textures and realistic relighting results from real-world images and noisy scanned geometry, demonstrating strong robustness to uncontrolled lighting and diverse inputs. We provide renderings under casual illumination conditions from two different views.

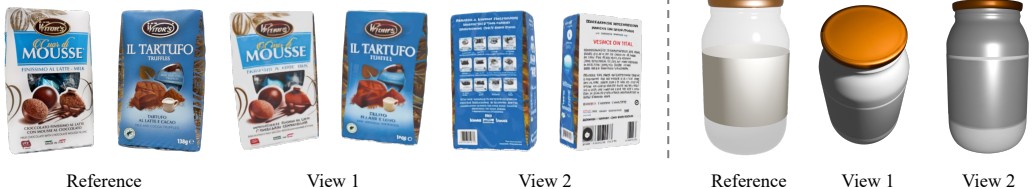

| Reference | View 1 | View 2 | | Reference | View 1 | View 2 |

Figure 14. **Failure Cases.** LumiTex faces several limitations. We provide the reference image and the generated textures rendered from two different views. First, the current resolution restricts the generation of fine details such as small printed text or specifications (left). Second, the model struggles with transparent materials due to the lack of alpha channel modeling (right).

**Training Details of Multi-View PBR Generation Transformer.** As introduced in Sec. 4.1 of the main paper, we initialize our model weights from FLUX.1-dev (BlackForestLabs, 2024). Then we adapt the model to a geometry-conditioned multi-view image generation generator by improving both multi-modal DiT and multi-view DiT as illustrated in Sec. 3.1. Specifically, to generate multi-view images of $N = 6$ views, for each view, we concatenate the latent features with geometry tokens and reference image tokens. These inputs are fused using a multi-modal DiT comprising $l_1 = 19$ double-stream transformer blocks. After fusion, we drop the geometry and reference tokens and concatenate the latents from all $N$ views. To encode view identity into each latent, we modify the factorized 3D Rotary Positional Embedding (3D RoPE), indexing each token by its space-view coordinate as $(t, i, j) = (0, h, w) + t \times (1, 0, o)$, where $t$ is the view index and $o$ is a fixed offset controlling inter-view separation. Finally, the multi-view DiT denoises the aggregated latents using $l_2 = 38$ single-stream transformer blocks. We train the model with the flow matching loss. The feature dimension is $C=3072$, and the token length is $L=1024$ for a resolution of $512 \times 512$. At each training timestep $t$, the model $G_\theta$ generates diffusion noise $G_\theta(\boldsymbol{I})$ for image latent $\boldsymbol{I}$, and the optimization loss $\mathcal{L}_{pbr}$ is calculated as:

$$\mathcal{L}_{pbr} = \mathbb{E}_t \left[ \sum_{i=1}^{N} \| G_\theta(\boldsymbol{I}_t^i) - \hat{\boldsymbol{I}}_t^i \|_2^2 \right], \tag{11}$$

where $I_t^i$ is the noisy latent of the ground truth shaded image from the $i$-th view. We first train the multi-view illumination context branch for 20,000 steps. Then we freeze their weights and train the PBR generation transformer for 20,000 steps. We utilize the prodigy optimizer (Mishchenko & Defazio, 2024) to self-adjust the learning rate and correct the bias. The $\beta_1$, $\beta_2$ are set to 0.9 and 0.999 respectively. During training, we use a batch size of 32 and apply a warmup phase of 2,000 steps. We apply gradient clipping at 1.0 and set the guidance scale to 1.0.

Benefiting from the rotary position embedding, our model could be trained with flexible aspect ratios (Esser et al., 2024). We first train our model with the image resolution $512 \times 512$. Then, we shift training to $768 \times 768$ for another 10,000 steps, and employ a timestep scheduler with a shift value $\alpha = 3.0$ (Esser et al., 2024).

**Training Details of LVSM-based Texture Inpainting Model.** The architecture of the texture inpainting model is adopted from the LVSM, which contains 24 full self-attention transformer layers with image patch size $p = 8$ and token dimension $d = 768$.

To train the texture inpainting model, for each object in the dataset, we randomly sample rendered images from $N = 6$ viewpoints along with their corresponding geometry images and camera parameters as conditions. The model is then tasked with generating the images for $M = 8$ additional randomly selected viewpoints. We train the model with the photometric and perceptual loss, the loss function is calculated as:

$$\mathcal{L}_{lvsm} = \sum_{i=1}^{M} \left( \text{MSE}(\hat{\boldsymbol{I}}_i, \boldsymbol{I}_i) + \text{LPIPS}(\hat{\boldsymbol{I}}_i, \boldsymbol{I}_i) \right), \qquad (12)$$

where $\hat{\boldsymbol{I}}_i$ is the predicted novel view images and $\boldsymbol{I}_i$ is the ground truth. We train our model for 10,000 steps on 8 GPUs using AdamW optimizer (Kingma, 2014) with a learning rate 4e-5. The $\beta_1$, $\beta_2$ are set to 0.9 and 0.95, respectively. During training, we use a batch size of 16 and apply a warmup phase of 1,000 steps. We also use a weight decay of 0.05 on all parameters except the weights of LayerNorm layers, following the original implementation of LVSM. During inference stage, we select $M = 18$ target views from the predefined dense set $\mathcal{V}$, which contains $K = 48$ candidate views.

**Inference.** We employ RMBG-2.0 (Zheng et al., 2024) to isolate the main foreground object from the generated image by removing the background. To prepare the 3D asset, the input mesh is first merged into a single connected component using Blender to avoid multiple UV lookups. We then use Xatlas to automatically generate UV parameterizations for material texture assignment. During inference, the guidance scale is set to 3.5. We implement a custom inverse renderer to map the generated materials (albedo, roughness, and metallic) from multiple views onto a unified UV texture space, producing ready-to-deploy 3D assets in GLB format with physically-based materials. To address view-dependent variations and ensure view consistency, we apply angle-weighted averaging across overlapping texels during projection. The resulting material maps are stored as UV textures with resolution $2048 \times 2048$, preserving the fine-grained details from the multi-view material maps. Our model requires approximately 1.5 minutes and 28GB of GPU memory to run inference at a per-view image resolution of $512 \times 512$, and around 3 minutes with 40GB memory at $768 \times 768$.

### A.3 LIMITATIONS

**Limited Resolution.** While LumiTex demonstrates strong performance in generating high-quality PBR textures, several limitations remain. First, generating multi-view images for PBR texture synthesis using diffusion transformer (DiT) models is computationally demanding. As a result, our current implementation is constrained to a per-view resolution of $768 \times 768$, which may be insufficient for extremely high-fidelity applications such as small printed text or specifications (see Fig. 14), film production or AAA game assets that require detailed textures at 4K or even higher resolution. Scaling to such high-resolution outputs presents challenges in both memory usage and inference time. Future work may explore improving the VAE compression ratio, optimizing transformer architecture (e.g., via sparse attention), or leveraging cached latents and multi-view parallelism to accelerate training and inference to improve quality.

**Transparency Modeling.** As shown in Fig. 14, our current pipeline does not support transparency modeling and thus fails to generate transparent materials such as glass, water, or translucent plastics. This limitation arises from the absence of an alpha channel or dedicated transmission parameter in the current material representation. Accurate modeling of these effects would require additional supervision, such as refraction-aware rendering or alpha/transmittance maps, which are not present in our training data. Future work could incorporate transparency channels and simulate light transmission effects to support a broader range of real-world materials.

### A.4 LLM USAGE DISCLOSURE

We used a Large Language Model (LLM), specifically GPT-4o, solely as a writing assistant to polish grammar, improve clarity, and enhance the fluency of the manuscript. The model was not involved in generating ideas, designing experiments, interpreting results, or contributing to the core research content. All scientific contributions, methodologies, and analyses were solely developed by the authors. The LLM's role was limited to editorial support and did not rise to the level of authorship or intellectual contribution.

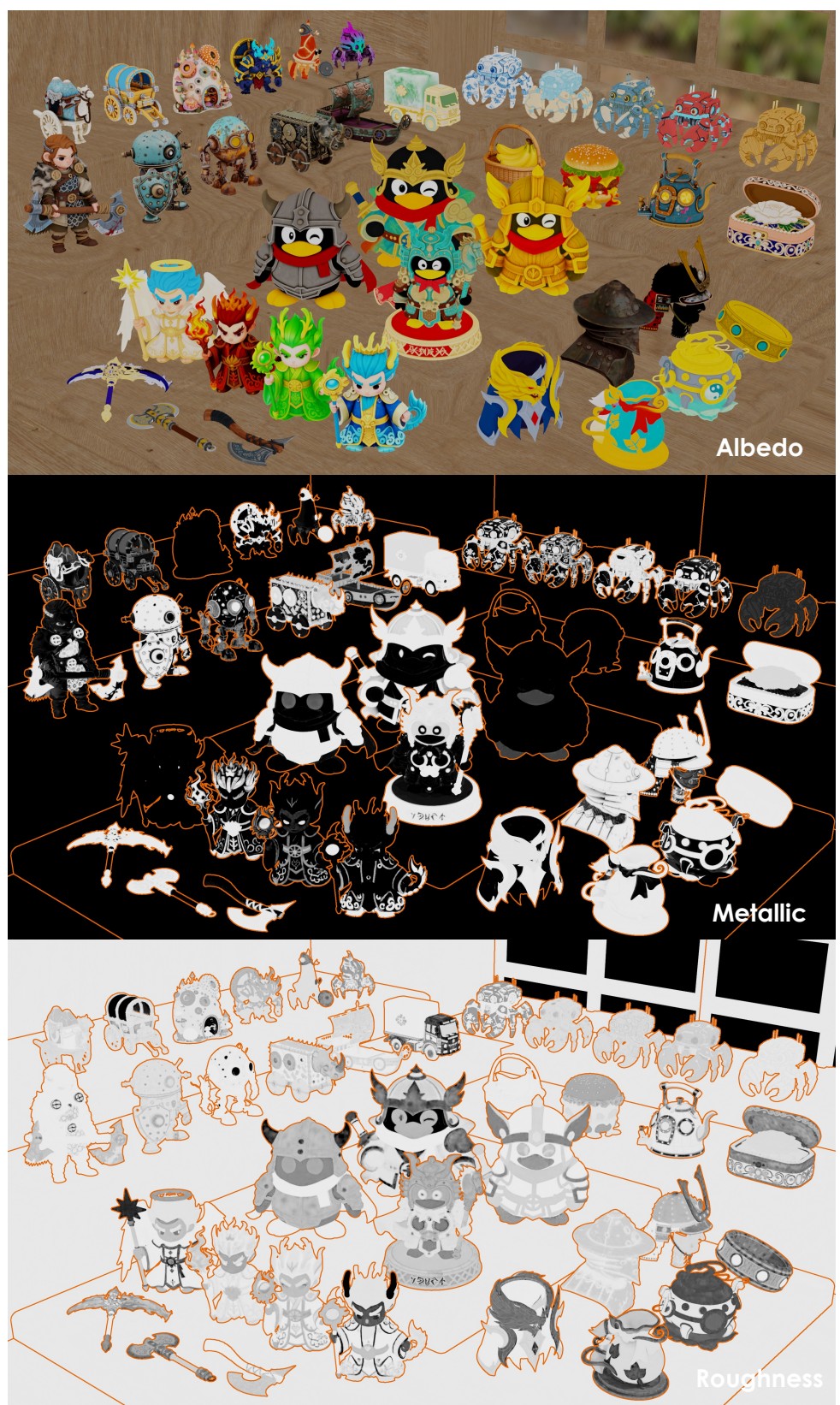

Figure 15. **G-Buffer Results of Fig. 1.** We report the full decomposed PBR results, containing 38 assets with both real and AI generated meshes and references. MR is outlined by silhouettes.

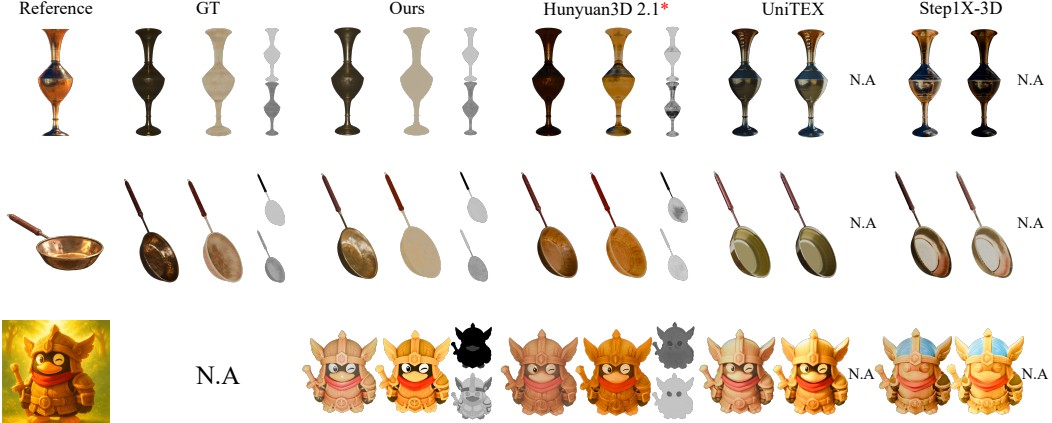

Figure 16. **Comparisons under Extreme Conditions.** We evaluate cases with highly reflective materials and strong backlighting. Our results (left: rendering, middle: albedo, top right: metallic, bottom right: roughness) avoid baked-in reflections and highlight artifacts, demonstrating greater robustness than existing approaches.

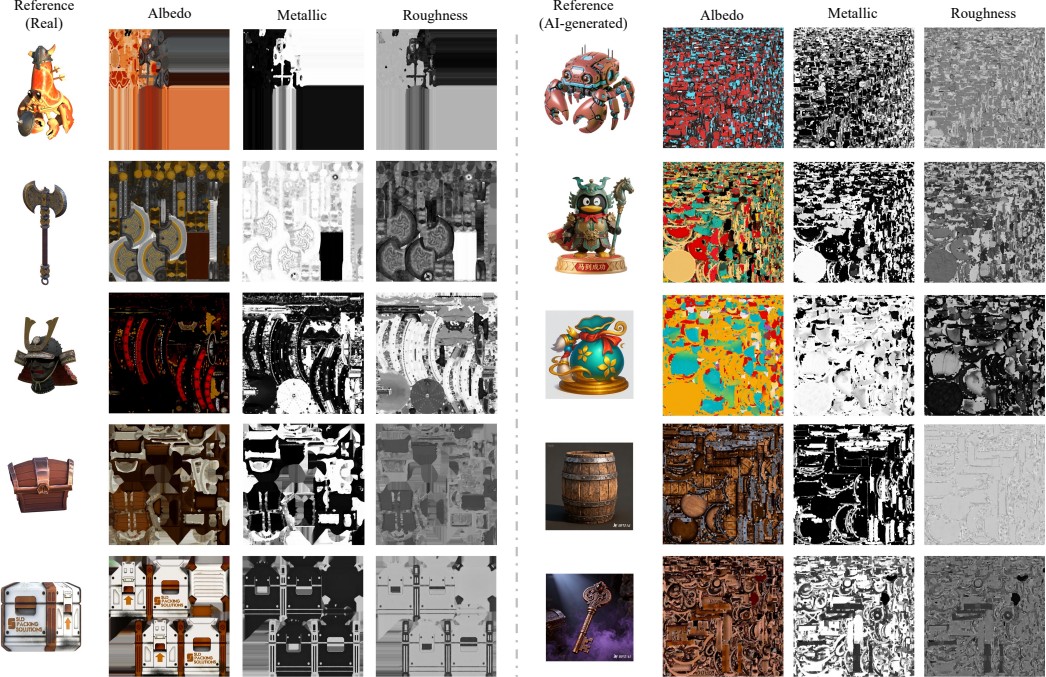

Figure 17. **Decomposed Material UVs for Fig. 10 and Fig. 11.** We present the decomposed material UV maps produced by our method. The left panel shows results generated from real rendered images, while the right panel shows results obtained from AI-generated reference images.

