# OpenReview forum: "LumiTex: Towards High-Fidelity PBR Texture Generation with Illumination Context"
_ICLR.cc/2026/Conference — ICLR 2026 Poster_

### Official Review · Reviewer_EhkN · 2025-10-29

**Soundness:** 4
**Presentation:** 4
**Contribution:** 3
**Rating:** 8
**Confidence:** 4

**Summary:**

This paper introduces LumiTex, a PBR texture generation model that achieves SoTA quality for PBR texture painting for 3D assets. LumiTex strengthens the connection between illumination shading and PBR material decomposition in the context of 3D generative models. The texture completion via LVSM has also proved to be very effective.

**Strengths:**

* The integration of illumination context implicitly through attention layers is a wise design. Additional illumination prior can effectively reduce shading ambiguity in PBR tasks.
* A joint model for multi-view, multi-channel PBR materials is a good attempt. The results demonstrate its effectiveness in reducing accumulation errors compared to prior multi-stage approaches.
* LVSM for texture completion performs very well in ensuring global consistency and seamlessness of the final texture maps.

**Weaknesses:**

I don’t find any major weakness point. Below are a few minor weaknesses points.
* It is unknown how MR is represented and decoded. Is MR in the orm convention or modeled separately? Is MR latents decoded with the same VAE used for regular RGB images?
* For the LVSM texture inpainting part, authors seem not to mention how decomposed PBR textures get inpainted. The example (Fig. 4) is on the rendered images.

**Questions:**

The LumiTex DiT is a native multi-view model, implying it should be capable of performing texture inpainting for additional views. I am curious why the authors opted for LVSM instead of leveraging the LumiTex DiT for this task and would appreciate some insights into this decision.

---

> ### Author Response · Authors · 2025-11-18
>
> We sincerely thank the reviewer for the detailed assessment and the strong recognition of our method. Your observations are insightful and have been very helpful in improving the technical clarity of our work. We hope the following responses fully address your questions.
>
> **Q1: The representation and decoding of MR**
>
> Yes, we represent MR maps in the ORM layout (occlusion–MR), with the occlusion channel set to a constant value. We employ the same VAE encoder–decoder architecture as used for standard RGB images, and empirical tests show that this encoding process is nearly lossless.
>
> **Q2: Texture inpainting using LVSM and design choice**
>
> **LVSM for PBR Inpainting.** The principle is the same for all 3-channel maps (also albedo and MR). Given the six generated albedo and MR images, we feed them separately into the geometry‑guided LVSM to infer two dense sets of views, i.e., the inpainting processes for albedo and MR are independent. We then back-project the six generated albedo images together with the inferred dense albedo views to UV space to obtain a complete albedo texture, and we apply the same procedure to MR.
>
> **Design Choice.** LVSM and LumiTex-DiT are both good at multi-view quality, but LVSM is a much more scalable, lightweight, and efficient choice. We hope our experience can provide some insights for this design.
>
> - **Quality:** Early in this work, we also experimented with using LumiTex‑DiT to synthesize additional views conditioned on six previously generated views. Concretely, we concatenate N condition views and M novel view latents to LumiTex‑DiT without finetuning. We found that the model can output views consistent with the condition, but the image quality drops when extremely dense views are requested (e.g., 6+18 views in our setup), likely due to the dense view being out of our training distribution. Instead, LVSM is known for its scalable architecture (only 24 fully self-attention layers) and strong zero-shot generalization, and it performs better on dense view generation.
> - **Efficiency:** More importantly, LVSM is much faster at inference speed given its lightweight architecture compared to DiT. We report the breakdown inference time of our pipeline without pipe acceleration:
>
>   | Method         | Params | View Res. | View Number | Time |
>   |----------------|----------|-------------|-----------------|-------|
>   | LumiTex-DiT | 12B       | 512^2       | 6 views          | 54s   |
>   | LVSM            | 171M    | 512^2       | 48 views        | 7s     |
>   | LumiTex-DiT | 12B       | 768^2       | 6 views          | 148s |
>   | LVSM            | 171M    | 768^2       | 48 views        | 9s     |
>
> - Our another insight to introduce the LVSM for inpainting is that LVSM achieves high-quality coverage with as few as six input views, as demonstrated in its original paper (Fig. 5 in LVSM). This matches exactly the number of views used in mainstream multi-view diffusion models, making it an efficient and well-aligned module that integrates naturally into our pipeline.
>
> Given its stability in quality and significant inference efficiency, we choose LVSM as our texture inpainting module. If you have any questions, we would be more than happy to discuss.

---

> ### Author Response · Authors · 2025-11-26
> **Follow-Up on Rebuttal Discussion**
>
> Dear Reviewer EhkN,
>
> We deeply appreciate your valuable feedback on our work that has helped us refine our work (we added LVSM inpainting details for PBR, Line 301-304), and value the opportunity to engage in further discussion to see if our response solves the concerns. We would greatly welcome any additional feedback or suggestions you may have.
>
> Thank you again for your devotion to the review. If all the concerns have been successfully addressed, we would appreciate it if you would reconsider the score accordingly.
>
> Best regards,
>
> Paper1297 Authors

---

### Official Review · Reviewer_RwEh · 2025-10-30

**Soundness:** 4
**Presentation:** 3
**Contribution:** 3
**Rating:** 8
**Confidence:** 4

**Summary:**

Authors propose an end-to-end framework that, given a reference image and a mesh, generates textures with PBR materials.

Main contributions:
- Multi-branch design (one for generation of shaded images - to capture illumination context).
- 2-stage training of multi-view illumination context branch and material branch
- Lightning-aware material attention mechanism (directly attending to shaded tokens instead of using explicit intermediate images or optimization techniques)

**Strengths:**

To my knowledge, this is the first approach that is able to generate close-to-true PBR materials, without noticeable baked reflections or highlights. Given strong quantitative and qualitative evaluation, and the importance and complexity of the texturing task, I consider this work to be significant to the field.

The main contributions and strengths of the paper are clearly demonstrated and ablated in section 4.5, namely:
- Separate branch for shaded images prediction
- Single-stage generation (no use of explicit intermediate shaded images)
- (!) Multi-branch design (instead of multi-channel generation of albedo and MR)
This was very insightful to learn, and well-supported by visuals in Fig 9b.

It was also demonstrated that the pipeline generalizes well to real-world scenes which is helpful in practical applications.

**Weaknesses:**

- Impact and novelty of geometry-guided inpainting module is limited, although this is not claimed as a main contribution of the paper.

**Questions:**

1. Please clarify the novelty of the inpainting module. Is the main contribution that you added geometry guidance to LVSM?

---

> ### Author Response · Authors · 2025-11-18
>
> We sincerely thank the reviewer for the positive assessment and strong recognition of our method. We would like to express our warmest appreciation for your support of our paper, fairness in evaluation, and clarity in identifying factual mistakes in other reviews. Here is our response to your question.
>
> **Novelty of the Inpainting Module**
>
> We acknowledge that integrating geometry conditions into LVSM alongside test-time training may appear technically simple. However, to the best of our knowledge, our method is the first work to repurpose a novel-view-synthesis (NVS) foundation model for the texture inpainting task. Prior texture-inpainting approaches (Paint3D, TexGen) operate strictly in UV space, thereby overlooking the rich semantic priors and diverse visual knowledge learned by large 2D foundation models -- priors that are well beyond the scale, diversity, and coverage of current 3D datasets.
>
> A second key insight behind our design is the observation that LVSM achieves high-quality coverage with as few as six input views, as demonstrated in its original paper (Fig. 5 in LVSM). This matches exactly the number of views used in mainstream multi-view diffusion models. As a result, LVSM provides an efficient, well-aligned, and high-fidelity inpainting module that integrates naturally into our pipeline, delivering significantly better completeness and global consistency compared to UV-space inpainting.
>
> ---
>
> We would also like to share some details of our early attempts in texture inpainting, which differ from the mainstream UV inpainting, and informed our decision to adopt LVSM.
>
> - **Generative 3D methods:** We experimented with training a point-cloud-based texturing model that colors unpainted 3D points conditioned on points colored by the six generated views. This approach often produces patchy or inconsistent results due to the limited semantic fidelity of point cloud representation relative to 2D images.
> - **Interpolation-based methods:** We also experimented with the texture inpainting module built upon optimization-based (e.g., MATCha [1]) or feed-forward sparse 3DGS methods (e.g., GS-LRM [2]) to fill the occluded regions from the six generated views. However, these approaches either struggle to fit the geometry due to the limited sparse views, or yield over-smoothed completions lacking high-frequency details since they do not create new information. This explicit 3D representation also inherently limits model flexibility.
>
> Although our final mechanism is simple, it outperforms UV inpainting, generative 3D methods, and interpolation-based approaches in detail preservation and surface coherence. We believe that leveraging an NVS foundation model for texture inpainting provides a promising new direction that effectively bridges 2D foundation models and high-quality texture completion tasks.
>
> References:
>
> [1] MAtCha Gaussians: Atlas of Charts for High-Quality Geometry and Photorealism From Sparse Views. CVPR 2025.
>
> [2] GS-LRM: Large Reconstruction Model for 3D Gaussian Splatting. ECCV 2024.

---

> ### Author Response · Authors · 2025-11-26
> **Follow-Up on Rebuttal Discussion**
>
> Dear Reviewer RwEh,
>
> We deeply appreciate your valuable feedback on our work that has helped us refine our work, and value the opportunity to engage in further discussion to see if our response solves the concerns. We would greatly welcome any additional feedback or suggestions you may have.
>
> Thank you again for your devotion to the review and for increasing the confidence!
>
> Best regards,
>
> Paper1297 Authors

---

### Official Review · Reviewer_yPWH · 2025-10-31

**Soundness:** 1
**Presentation:** 1
**Contribution:** 2
**Rating:** 2
**Confidence:** 4

**Summary:**

The paper proposes a method to generate PBR textures for given objects. The core idea are two stages: from an input mesh + reference image, the goal is to generate N view-consistent PBR images. Here, the core idea is essentially a multi-view image generator with several conditions (they authors refer to this as an illumination-consistent base model); this is then frozen and a material branch is trained for the PBR part. The second stage is a geometry-guided LVSM to synthesize more viewpoints. This is a texture in-painting strategy based on LVSM which generates the extended views, hence, more complete textures. Training is done from 92K objects from Objaverse and Objaverse-XL – for each object, 30 views are rendered (albedo, metallic, roughness, and HDR images). The base model is FLUX.1-dev.

However, to be honest, the main technical exposition is quite confusing and it’s not easy to follow the exact details of the multi-view PBR generator (more details below).

**Strengths:**

- The authors tackle an important problem.

- The renderings of the shaded outputs look nice.

- I appreciate the re-lighting results in the video.

**Weaknesses:**

The presentation is confusing, and I'm having trouble understanding several of the technical details:

- The introduction mostly pitches the features but a coherent description of the core idea; e.g., how does the multi-view shaded image generator work is somewhat omitted – this makes it hard to read (e.g., first need to read the whole main section and even some of the results to understand which base models they were using)

- Fig 3 is a pipeline but the description of the multi-view illumination-consistent base model is missing, and the pipeline flow is confusing (e.g., where does the input mesh go, where do the reference images come into play).

- I’m confused about the term multi-modal DIT. Seems all input here are images… what are the modes you are referring to here?

- In the video, while the re-lighting looks great, I would’ve loved to see the actual PBR materials rather than the shaded versions. Also the shading in the video seems exhibit some temporal unstable artifacts which is confusing given that the underlying rendering should be just a mesh + PBR texture = this should be visualized.

- The main results are in Fig 6 in the main paper (the majority of visuals does not show the actual PBR results but the shaded versions). Unfortunately, this looks not that impressive. E.g., I would’ve loved to see the PBR textures of the objects in Fig 1 or Fig 5 instead of the shaded outputs.

There is a general confusing claim to  PBR textures but then consider environment lighting which would still mean that scene specific context is baked in. This is contrary to the definition of a PBR texture.

Figure 2 albedo looks poor. These results look shaded as there is lighting baked in – this is unfortunately not a PBR image / texture. Am I missing something here?

**Questions:**

See above in the weakness sections.

---

> ### Author Response · Authors · 2025-11-18
> **Official Comment by Authors (1/2)**
>
> Thank you for your thoughtful comments and valuable suggestions. We will revise our paper based on your feedback. Here are our responses to your comments:
>
> **Q1: The introduction mostly pitches the features but a coherent description of the core idea; e.g., how does the multi-view shaded image generator work is somewhat omitted.**
>
> We appreciate the reviewer's feedback. The central idea of our work is a novel multi-branch architecture combined with lighting-aware attention, which enables a generative model to bridge illumination cues and PBR material properties more effectively. To keep the introduction focused, we primarily highlighted the limitations of existing PBR-generation paradigms and how our approach addresses them.
>
> Our multi-view shaded image generator is a traditional multi-view diffusion model that enforces cross-view consistency via the multi-view attention. Different from text-to-image diffusion models, a multi-view diffusion model **jointly reasons about shared 3D structure and appearance across all views**, rather than generating each view independently. Specifically, they extend the self-attention of text-to-image diffusion models to include all pixels across multi-view images. Let $f^{in}$ denotes the input of the attention block, including geometry condition and latent tokens. The dense multi-view self-attention extends $f^{in}$ from the view itself to the concatenated feature sequence from n views, as in eq. 5 in our paper. During training, the model is supervised by multiple renderings of the same asset, enabling it to learn consistent geometric and shading cues.
>
> We appreciate the reviewer's suggestion, and with the additional page available, we will include a brief schematic to make the writing more self-contained and accessible.
>
> **Q2: The pipeline flow of Fig. 3 is confusing (e.g., where does the input mesh go, where do the reference images come into play), the description of the multi-view illumination-consistent base model is missing.**
>
> We thank the reviewer for this suggestion, and we have revised this figure based on your feedback.  Here are our responses to your comments:
>
> - **Pipeline flow.** Similar to MV-Adapter, UniTex, Hunyuan3D-2.1, the input mesh is used to render geometry conditions (i.e., multi-view normal maps and canonical coordinate maps) in the left yellow block. Then, these geometry condition maps are added and encoded with VAE and the embedder layers to derive condition tokens (eq. 1, the yellow rectangle). As shown in the top purple block of the figure, the reference image is encoded by DINOv2 and VAE and concatenated to the image feature token after the embedder (eq. 2, the purple rectangle).
>
>     Then, for the material attention layers, these tokens attend the multi-view illumination branches to derive the shaded keys and values (eq. 5) and perform attention with the albedo and MR queries derived from the material branch. Finally, the latents are decoded to multi-view albedo images and MR images.
>
> - **Description of the multi-view illumination-consistent base model.** Also, similar to texture generation methods (MV-Adapter, UniTex, Hunyuan3D-2.1), this model (the multi-view illumination branch) outputs multiple views given the input images. The training of this model is to supervise the outputs using multi-view images rendered from 3D assets under GT environment lighting.
>
> **Q3: I’m confused about the term multi-modal DIT. Seems all input here are images… what are the modes you are referring to here?**
>
> - As shown in eq. (1, 2, 3). The modes of inputs are RGB images (normal/CCM, reference; dimension=3), DINOv2 latents (dimension=768), and learnable embeddings (dimension=4096). Therefore, we refer to this module as a multi-modal DiT.
>
> **Q4: The majority of visuals does not show the actual PBR results but the shaded versions, and would love to see PBR textures.**
>
> We kindly direct the reviewer to Fig. 6, Fig. 10, and Fig. 11, which present the full decomposed PBR results (albedo, normal, roughness, and metallic). These figures are specifically designed to showcase the intrinsic material maps produced by our method.
>
> Fig. 5 shows the shaded texture results, as they provide a comprehensive and fair comparison within the broader texture-generation domain. Many competing methods focus solely on shaded appearance and do not output decomposed PBR channels. Therefore, we report the shaded results they make available to ensure consistency across baselines.
>
> Additionally, we further report the full decomposed PBR results of Fig. 1, which contains 38 assets generated by our methods with both real and AI-generated meshes and references in **Fig. 15**, offering a comprehensive visualization of our method’s outputs.

---

> ### Author Response · Authors · 2025-11-18
> **Official Comment by Authors (2/2)**
>
> **Q5: There is a general confusing claim to PBR textures but then consider environment lighting which would still mean that scene specific context is baked in.**
>
> Our motivation for integrating lighting context into the PBR framework is to enable the model to learn intrinsic material properties rather than inadvertently absorbing illumination effects into the generated textures. In the real rendering process, **material × lighting → appearance**, where the appearance comes from the user-provided reference image. Therefore, providing the model with explicit illumination cues (lighting) allows it to better disentangle shading from the underlying albedo, roughness, or metallic properties.
>
> Without such lighting context, the diffusion model is more prone to **struggling with MR decoupling**, which we also validate in our ablation: removing the multi-view illumination branch leads to a noticeably plastic appearance (Fig. 9).
>
> Compared to SoTA two-stage pipelines that rely on intermediate shaded predictions, which often lead to error accumulation and entangled shading priors, our approach integrates illumination cues directly through attention to shaded tokens within a unified diffusion transformer. This end-to-end design avoids imperfect intermediate supervision and improves material–lighting separation, as reflected in our quantitative and qualitative results.
>
> **Q6: Figure 2 albedo looks poor. These results look shaded as there is lighting baked in – this is unfortunately not a PBR image / texture.**
>
> We appreciate the reviewer’s observation and want to clarify that the albedo results in Fig. 2(c) are the closest to physically correct, lighting-free PBR materials among the compared SoTA methods. As shown in the figure, both SoTA PBR texture generation pipelines, **(a) IDArb** and **(b) Hunyuan3D-2.1**, exhibit visible baked-in lighting, including residual shading and highlight imprints. In contrast, our method produces cleaner and more illumination-invariant albedo.

---

> > ### Comment · Reviewer_yPWH · 2025-11-22
> >
> > First of all, I appreciate the reviewer's quick response to my questions. Please see my comments below.
> >
> > Q1: The method in itself is relatively straightforward; however, it requires a careful read-through of the main method section to fully understand it. My major complaint is about the introduction which is confusing and makes the paper difficult to follow. In a nutshell, it's obfuscating the core part of the multi-view model and I would advise to improve it.
> >
> > Q2: Thanks for revising the figure; it would be great if was implemented in the final paper.
> >
> > Q3: I'm aware of the inputs, but disagree with the definition of multi-modal. In my opinion, this would refer to different domains such as text vs audio vs visual input. Hence, the claim here is somewhat misleading.
> >
> > Q4: These (Fig. 6, 10, 11) are rendered materials in a G-Buffer but not textures. I would've loved to see the PBR textures instead. This is quite relevant as the core contribution is a multi-view model that produces consistent textures rather than per-view material maps.
> >
> > Q5: thanks for clarifying - not sure if it makes sense in practice but I can see where you're coming from.
> >
> > Q6: yes, methods that start from pre-trained real-world priors tend to bake in lighting which we can see here; however, those that train directly from PBR data do not. I do see the better generalizability advantage though. Obviously, the rendered visuals are nice but I was not overwhelmed by the decomposition and I don't see it close to physically correct.

---

> ### Author Response · Authors · 2025-11-23
>
> We thank the reviewer’s timely feedback. We noticed your emphasis on the “physically correct” “PBR textures” (UVs), and we would like to first clarify an important distinction between **two subdomains of PBR texture research**, to avoid potential misunderstanding:
>
> 1. **PBR map creation** (e.g., Material Palette, MaterialPicker, Chord):
>
>     These physically accurate oriented methods recover BRDFs (albedo, metallic, roughness, etc.) directly in UV or texture space, often assuming clean inputs, structured patterns, and controlled setups. While physically accurate, this approach generally exhibits limited generalizability when applied to casual meshes or images.
> 2. **PBR texturing (for a mesh) via multi-view synthesis** (e.g., DreamMat, MaterialMVP, Hunyuan3D series):
>
>     These methods aim to generate PBR textures for arbitrary 3D meshes and arbitrary images, where reference alignment, texture completeness, and illumination disentanglement are primary challenges.
>
> Our work lies in the second subdomain, which focuses on complete texturing of 3D assets under complex appearance and illumination conditions. It’s worth-noting that **no existing method simultaneously solves both subdomains** -- i.e., achieving perfect physical plausibility at the map-creation level and strong generalizability for arbitrary mesh texturing. These remain partially overlapping yet fundamentally different research directions.
>
> ---
>
> Here are our responses to the remaining questions:
>
> **Q1: It's obfuscating the core part of the multi-view model in the intro and I would advise to improve it.**
>
> Thanks for your suggestion. We have revised our manuscript (Line 53-63) to add a paragraph to specify the research question we study (PBR texturing for a mesh) and the preliminary (multi-view model) of our work.
>
> **Q3: I'm aware of the inputs, but disagree with the definition of multi-modal, which would refer to different domains such as text vs audio vs visual input.**
>
> We respectfully correct the definition of `multi-modal`: it’s not limited to domains like text/audio/visual. The multi-modal could also refer to combining multiple **visual channels** without text, as we used in the paper. For example, in the PBR literature, Chord [1], IntrinsicEdit [2], and GANtlitz [3] explicitly treat basecolor, normal, metallic (specular) as multiple modalities. [4] and [5] also explicitly named RGB and depth as “multi-modal”.
>
> References:
>
> [1] Chord: Chain of Rendering Decomposition for PBR Material Estimation from Generated Texture Images. SIGGRAPH Asia 2025.
>
> [2] IntrinsicEdit: Precise generative image manipulation in intrinsic space. SIGGRAPH 2025.
>
> [3] GANtlitz: Ultra High Resolution Generative Model for Multi-Modal Face Textures. Eurographics 2024.
>
> [4] GeoWizard: Unleashing the Diffusion Priors for 3D Geometry Estimation from a Single Image. ECCV 2024.
>
> [5] Multimodal Material Segmentation. CVPR 2022.
>
> **Q4: I would've loved to see the PBR textures instead.**
>
> We provide the PBR textures (UV) results of Fig. 10, 11 in Fig. 17. As noted in the implementation details, our UV is parameterized based on XAtlas.
>
> We would also like to kindly note that, due to the nature of multi-view texture synthesis, the rendered appearance serves as the primary evaluation target while the UV maps are relatively auxiliary -- this is standard practice in the literature (e.g., DreamMat, MaterialMVP, Hunyuan3D), which is different from the UV-direct approaches.
>
> **Q6: Methods start from real-world priors have better generalizability advantage but inferior PBR accuracy than directly train from PBR data.**
>
> We agree with this point of view and these represent **two orthogonal research tracks**. As clarified above, our work primarily targets *PBR texturing (for a mesh) via multi-view synthesis*. Within this domain, we kindly note that **our results (Table 1) achieve ~35% improvement over SoTA open-source baselines** on key metrics, and **~20% improvement over methods trained on private datasets**. We hope it is clear that this work aims to advance the particular direction, rather than simultaneously solve both research tracks, which involve fundamentally different goals and data assumptions.
>
> In this subdomain, achieving more physically accurate materials is inherently more challenging than UV-direct approaches. It's about trade-offs between quality and generalizability. Nevertheless, multi-view texture synthesis remains the mainstream for mesh texturing, and our work represents a milestone that provides a strong SoTA solution in both texture quality and texture completeness.
>
> ---
>
> Lastly, thank you for the timely feedback. We are glad the concerns **Q2,5** are addressed, and we hope our response also addresses the remaining points. Given the clarified research scope and the gap our work aims to address, we hope the reviewer could re-evaluate the contribution of our work, and we would appreciate it if you would reconsider the score accordingly.

---

> ### Author Response · Authors · 2025-11-26
> **Follow-Up on Rebuttal Discussion**
>
> Dear Reviewer yPWH,
>
> We deeply appreciate your valuable feedback on our work that has helped us refine our work, and value the opportunity to engage in further discussion to see if our response solves the concerns. We would greatly welcome any additional feedback or suggestions you may have.
>
> Thank you again for your devotion to the review. If all the concerns have been successfully addressed, we would appreciate it if you would reconsider the score accordingly.
>
> Best regards,
>
> Paper1297 Authors

---

### Official Review · Reviewer_t6H4 · 2025-11-01

**Soundness:** 3
**Presentation:** 3
**Contribution:** 2
**Rating:** 2
**Confidence:** 4

**Summary:**

The paper proposes LumiTex, a PBR texture generation framework that addresses challenges like material decomposition with limited illumination cues and seamless texture completion. LumiTex combines a multi-branch generation scheme, a lighting-aware material attention mechanism, and a geometry-guided inpainting model to enhance texture quality, realism, and consistency across views. Extensive evaluations are delivered.

**Strengths:**

- First of all, the work proposes the multi-branch generation design and the lighting-aware attention mechanism offers a novel way of disentangling albedo and metallic-roughness (MR) while integrating illumination context.


- Quantitative results (FID, CMMD, LPIPS) and qualitative evaluations indicate that LumiTex achieves competitive or superior performance compared to existing methods, particularly in terms of texture quality and relighting fidelity.

- The authors conducted a wide range of experiments, including comparisons to both open-source and commercial systems, as well as a user study on texture quality, demonstrating LumiTex’s practical advantages in real-world applications.

**Weaknesses:**

- While the framework employs multi-branch generation and illumination context, similar ideas have already been explored in other recent works. For example, the idea of using lighting priors for material generation is not very novel. The originality of LumiTex comes into question because the combination of multi-view consistency and lighting-guided material attention don't significantly advance the state of the art in a groundbreaking way.

- The method is computationally intensive and limited to generating textures at 768×768 resolution, which severely restricts its application in high-end industries that require 4K or 8K resolution for detailed textures (IMO this is more useful for texture generation for gaming applications or AAA filming). Although the authors propose potential avenues for scaling (e.g., multi-resolution models), scalability remains an unresolved bottleneck. Real-time applications (such as interactive design in AR/VR) would be severely constrained by the current training time of 106 GPU days. The paper does not adequately explore solutions to this scalability issue.


- The lack of support for transparent materials. Transparent materials, such as glass, water, and liquids, are commonly required in real-world 3D rendering, especially in architectural visualization and product design. The authors don’t provide a solid argument for why these materials are left out or when this limitation might be addressed. There already exist many works that investigate transparent image / video generation.

- LumiTex doesn’t show sufficient results in handling complex reflective materials or subsurface scattering. These properties are common in materials such as skin, water, and polished metals, which are common in visual effects and games.

- The generalization to novel inputs seems to be limited. While the paper demonstrates robustness with real-world scanned meshes, the model still relies heavily on the type of training data (from Objaverse and Objaverse-XL). The authors did not provide convincing results showing how well LumiTex generalizes to extremely diverse or out-of-distribution inputs. This is a key issue for any system claiming real-world applicability in fields such as gaming or film, where content varies vastly.

- The paper presents failure cases (such as small printed text or transparent materials), but it does not delve deeply into why these failures occur or how they might be addressed in future work. For instance, the lack of alpha channel modeling for transparent materials is acknowledged, but the paper does not offer a roadmap for incorporating transparency modeling or refraction effects, making it seem like a major limitation without any plans for resolution.

- Despite claims of robustness under various illuminations, the framework’s real-world reliability under extreme lighting scenarios (e.g., highly reflective surfaces, strong backlighting) is not well-documented. The existing results are focused on typical lighting conditions, and there is a risk that the model might fail in edge cases involving extreme or uncontrolled lighting, common in cinematic and interactive applications. Without those, the contribution of this paper is doubtful.

**Questions:**

I would be grateful if the authors could address the above mentioned issues in weakness section and other reviewers' concerns.

---

> ### Comment · Reviewer_EhkN · 2025-11-14
> **Review the review ;)**
>
> After seeing the new post from Reviewer RwEh, I also got interested to investigate the reason for the low rating from the other reviewers.
>
> Although there are many weaknesses listed here, but many points are off the main topic of the paper, or are limitations already stated  in the paper. (skeptical LLM-ish)
>
> > "Real-time applications (such as interactive design in AR/VR) would be severely constrained by the current training time of 106 GPU days"
>
> Hmmm, the proposed method is a feed-forward model. Training is a one-time cost, the inference in time can be way faster.

---

> ### Author Response · Authors · 2025-11-18
> **Official Comment by Authors (1/2)**
>
> Thank you for your comments. Here are our responses to your comments:
>
> **Q1: Using lighting priors for material generation is not very novel.**
>
> We agree that the relationship between lighting and materials has been explored for many years, as it lies at the core of inverse graphics. However, **effectively integrating illumination context into a generative model remains non-trivial**. Prior works typically incorporate lighting priors through (1) explicit optimization (e.g., DreamMat), (2) handcrafted constraints (e.g., Hunyuan3D-2.1), or (3) staged pipelines that rely on imperfect shaded intermediates (e.g., MuMA, Seed3D). As discussed in the paper, these designs often introduce saturation artifacts, illumination leakage, and error accumulation arising from noisy intermediate shading.
>
> In contrast, our method introduces **a novel multi-branch design with an implicit lighting-aware attention mechanism**, which, unlike prior works, conditions the material branch on shaded tokens without relying on explicit shaded intermediates. This design avoids the error accumulation common in the 2-stage design and effectively mitigates the data imbalance (high-quality MR data is scarce) problem in the open-source dataset.
>
> Reviewers EhkN and RwEh also recognized this as a strength, noting that *“the integration of illumination context implicitly through attention layers is a wise design”* and *“a very insightful multi-branch design.”*
>
> **Q2: Lighting-guided material attention doesn't significantly advance the state of the art in a groundbreaking way.**
>
> We respectfully clarify that the impact of our lighting-guided material attention is not incremental, but substantial and empirically verified. As shown in Table 1, our method **delivers >20% improvements across FID, CLIP-FID, CMMD, and LPIPS on texture evaluation** -- gains far exceeding what would typically be attributed to a minor architectural change. Even when compared with strong commercial systems (Meshy-5, TripoAI v2.5) and open-source models trained on private datasets (Hunyuan3D-2.1), our approach remains competitive and superior across nearly all metrics.
>
> This assessment is also supported by Reviewers EhkN and RwEh, who note that: “LumiTex achieves SoTA quality for PBR generation, the metrics (e.g., FID) in tables 1 and 2 show strong evidence, and it is able to generate close-to-true PBR materials.”
>
> **Q3: Real-time applications (such as interactive design in AR/VR) would be severely constrained by the current training time of 106 GPU days.**
>
> We respectfully clarify that **training time does not limit real-time or interactive applications, as training is performed once on GPU clusters, not on AR/VR or consumer devices.** We are therefore confused why real-time usage would be “severely constrained” by training cost.
>
> As Reviewers RwEh and EhkN also noted, our training time is well within the norm for current large generative models. For reference, the latest SoTA PBR-generation system Hunyuan3D-2.1 (MaterialMVP + private datasets, ICCV 2025) reports 180 GPU-days, which is 1.8× our training cost.
>
> Our model, like all texture-generation systems, is trained once and then deployed for inference on consumer GPUs. Our inference speed, without any engineering optimizations, is 1.5 minutes at a projection resolution of 512² and 3 minutes at 768² (Line 1052), providing practical consumer-grade performance suitable for interactive design workflows, which is competitive with other commercial systems.
>
> **Q4: The method is limited to generating textures at 768×768 resolution, which severely restricts its application in high-end industries that require 4K or 8K resolution for detailed textures.**
>
> We respectfully clarify that 768×768 is only the intermediate per-view image resolution, not the final texture resolution. As stated in the paper (Line 1050), our experiments already use **2048x2048** textures, and the pipeline naturally supports higher resolutions such as 4K or 8K. Notably, the final texture resolution is determined by the UV unprojection step rather than the 768×768 per-view rendering resolution. Our current setup aggregates 768×768×(6+18) ≈ 14M rendered pixels into the final texture map, providing sufficient coverage and detail for high-resolution outputs.  Reviewer RwEh also noted that this should not be considered a limitation of our approach.

---

> ### Author Response · Authors · 2025-11-18
> **Official Comment by Authors (2/2)**
>
> **Q5: Lack of support for transparent materials, alpha channel modeling, and complex subsurface scattering.**
>
> We would like to emphasize that **these points lie outside the scope** of our paper, as also noted by Reviewers RwEh and EhkN. Current state-of-the-art PBR and texture generation methods, such as SyncMVD (SIGGRAPH Asia 2024) [1], MV-Adapter (ICCV 2025) [2], DreamMat (SIGGRAPH 2024) [3], Hunyuan3D-2.1 / MaterialMVP (ICCV 2025) [4], and SuperMat (ICCV 2025) [5], **also do not support transparency, alpha channels, or complex subsurface scattering**. These effects are extremely underrepresented in open-source datasets. For example, Objaverse includes **fewer than 1%** assets with alpha channels, making them unsuitable targets for generative training.
>
> We have clearly acknowledged the lack of the alpha channel as a limitation in our paper (Line 1065). To the best of our knowledge, **no existing general generative method can robustly handle all transparent materials** or full BSDF modeling in an end-to-end framework. Given the scope and space constraints of a 9-page paper, it is also unrealistic and misaligned with the current literature to expect a method to cover all optical effect topics.
>
> Our work focuses on high-quality Albedo+MR PBR generation, and we believe this focus is appropriate and consistent with the scope of contemporary texture-generation research.
>
> **Q6: The model still relies heavily on the type of training data (from Objaverse and Objaverse-XL). The authors did not provide convincing results showing how well LumiTex generalizes to extremely diverse or out-of-distribution inputs.**
>
> We appreciate the reviewer highlighting the importance of evaluating generalization. We would like to clarify that **our experiments already include extensive out-of-distribution (OOD) testing across both input meshes and reference images**. First, as described in Sec. 4.2 (Line 376), we evaluate artist-created and AI-generated meshes originating from Sketchfab and GitHub assets that are explicitly excluded from the training data to ensure OOD coverage. Second, we demonstrate robustness to diverse reference images, including both real renderings and AI-generated inputs, as shown in Fig. 10 and Fig. 11.
>
> **Q7: Reliability under highly reflective surfaces and strong backlighting is not well-documented.**
>
> Regarding extreme illumination scenarios, we agree that this is an important aspect. To further strengthen the evaluation, we have added additional comparisons featuring **highly reflective materials** and **strong backlighting conditions**, as shown in Fig. 16. These results demonstrate that LumiTex maintains high-quality relightable materials even under challenging illumination conditions, supporting the claim that our model generalizes well beyond the distribution of Objaverse/Objaverse-XL. Notably, baseline methods fail in these cases because they lack explicit modeling of illumination context and therefore tend to bake severe shading artifacts into the materials.
>
> References:
>
> [1] Text-Guided Texturing by Synchronized Multi-View Diffusion. SIGGRAPH Asia 2024.
>
> [2] MV-Adapter: Multi-view Consistent Image Generation Made Easy. ICCV 2025.
>
> [3] DreamMat: High-quality PBR Material Generation with Geometry- and Light-aware Diffusion Models. SIGGRAPH 2024.
>
> [4] MaterialMVP: Illumination-Invariant Material Generation via Multi-view PBR Diffusion. ICCV 2025.
>
> [5] SuperMat: Physically Consistent PBR Material Estimation at Interactive Rates. ICCV 2025.

---

> ### Author Response · Authors · 2025-11-26
> **Follow-Up on Rebuttal Discussion**
>
> Dear Reviewer t6H4,
>
> We deeply appreciate your valuable feedback on our work that has helped us refine our work, and value the opportunity to engage in further discussion to see if our response solves the concerns. We would greatly welcome any additional feedback or suggestions you may have.
>
> Thank you again for your devotion to the review. If all the concerns have been successfully addressed, we would appreciate it if you would reconsider the score accordingly.
>
> Best regards,
>
> Paper1297 Authors

---

### Comment · Reviewer_RwEh · 2025-11-12
**Defending the "Accept" rating**

Given that I think that this paper is a strong contribution, I would like to defend my "Accept" rating, specifically by highlighting the critical flaws in "Reject" reviews below.

## Reviewer t6H4
> and limited to generating textures at 768×768 resolution

The method is not limited to generating textures at this resolution - it generated rendered shaded and materials images at this resolution. After subsequent LVSM module they unproject it into textures of higher resolution (theoretically unlimited). This is not a limitation of this method.

> Real-time applications (such as interactive design in AR/VR) would be severely constrained by the current training time of 106 GPU days.

106 GPU-days for a foundational materials generation model is not a lot (13 days in single-node 8-GPU training time) by today's industry standards. Also
> real-time applications (such as interactive design in AR/VR)

it is clear that the model will not be trained in "real-time" and certainly not on the devices common AR/VR industry folks use. It's trained once but then inferred on demand on consumer GPUs. So I do not agree these applications should be "severely constrained" by this training time.

> Transparent materials, such as glass, water, and liquids, are commonly required in real-world 3D rendering, especially in architectural visualization and product design

I would argue that transparent materials support is a niche task, rather than being "commonly required". Albedo + MR is much more commonly used and covers most use cases. Authors correctly list transparency as a limitation, so I do not consider this a critical weakness.

> the lack of alpha channel modeling for transparent materials is acknowledged, but the paper does not offer a roadmap for incorporating transparency

I think it's an additional line of work and not a core of this paper.

> The originality of LumiTex comes into question because the combination of multi-view consistency and lighting-guided material attention don't significantly advance the state of the art in a groundbreaking way.

The metrics (e.g. FID) in tables 1 and 2 shows strong evidence that it does advance the state-of-the art.

## Reviewer yPWH
> the pipeline flow is confusing (e.g., where does the input mesh go, where do the reference images come into play)
> how does the multi-view shaded image generator work is somewhat omitted

With respect, I would argue that you might have not understood the paper. I would advise lowering the confidence threshold in this case.

> the majority of visuals does not show the actual PBR results but the shaded versions

Please see Figure 10 and Figure 11.

> I’m confused about the term multi-modal DIT

I think authors mean this:
> The MM-T is designed to integrate geometry information, reference appearance, and material semantics (albedo or MR) for each view.

DINO features is not an RGB image, also camera view dirs are not an RGB image (although a tensor with dimensionality of 3). This is what authors likely mean by the term multi-modal.

## Conclusion
I think the paper contributions are clear and sufficient for an acceptance, and above mentioned reviews contain significant flaws. I further ask the reviewers to lower their confidence rating, given these issues were highlighted.

---

### Author Response · Authors · 2025-11-18
**General Response**

Dear ACs and Reviewers,

Thank you very much for your dedication, support, and insightful feedback.

PBR texture generation is a complex task, and achieving state-of-the-art performance against both open-source methods (trained on private datasets) and leading commercial methods is quite challenging. We are grateful for the reviewers' recognition of our work's strengths:

- **Novel pipeline design:** Reviewers noted that our multi-branch design "offers a novel way" for PBR texture generation and is "very insightful to learn" (t6H4, RwEh, EhkN), and our light-aware material attention that directly attends to shaded tokens "is a wise design" (RwEh, EhkN).
- **Effective Demonstration:** Reviewer highlighted that our single-stage design effectively avoids error accumulation inherent to previous multi-stage pipelines, and that the integration of illumination context in one branch can "effectively reduce shading ambiguity" (RwEh, EhkN). They also recognized that "LVSM for texture completion performs very well" (EhkN).
- **Strong Experimental Results:** Reviewers praised that our approach is "the first approach that is able to generate close-to-true PBR materials" (RwEh), and our "wide range of experiments" provides "strong evidence" compared to previous approaches (t6H4, RwEh, EhkN). They also commended the model's ability to generalize well to real-world scenarios (t6H4, RwEh).

We have reviewed all the comments, addressed all questions, and provided additional experimental results. All revisions are highlighted in **red** in the updated version, and we summarize the revisions we made:

- **Additional Experimental Results (yPWH; Line 834):** Added PBR decomposition results of 38 assets for real and AI-generated meshes and references in Fig. 15 and Appendix A.1.
- **Robustness under Extreme Conditions (t6H4; Line 837):** Added results under extreme conditions in Fig. 16, Appendix A.1.
- **Texture Resolution (t6H4; Line 1052, 1060):** Clarified the texture and per-view image resolution.
- **Figure Clarity (yPWH):** Added more detailed annotations of our pipeline in Fig. 3.
- **Preliminary in Intro (yPWH; Line 53-63):** Added the preliminary of the multi-view model to make our intro more self-contained.
- **UV visualization (yPWH):** Added the UV results of our PBR in Fig. 17.
- **Inpainting Details (EhkN; Line 301-304):** Added LVSM inpainting details for PBR.

**Request for Feedback**

We respectfully invite the reviewers to carefully evaluate our revisions and the individual responses provided. We are more than willing to address any remaining questions or concerns. If our responses and the additional results sufficiently address your feedback, we would appreciate your consideration of increasing your scores. We sincerely appreciate your thoughtful engagement and constructive suggestions, which have been instrumental in enhancing the quality of this work.

Best regards,

Authors of Submission 1297

---

### Public Comment · ~Shuhui_Yang2 · 2025-11-28
**Open Reviewers**

Interesting, I couldn't help noticing that this paper features so many of my own carefully selected​ cases for MaterialMVP and Hunyuan3D platform and technical reports.

---

### Author Response · Authors · 2025-12-02
**Rebuttal Summary and Clarifications**

Dear reviewers and ACs:

We are writing to provide a summary of our rebuttal interaction and a summary of our work.

**Summary of Rebuttal Interaction**

Before the discussion period was interrupted, two reviewers provided updated feedback:

Reviewer yPWH expressed appreciation that our revised figures and clarified motivation for incorporating illumination context made the overall pipeline clearer, and advised to polish the introduction part with the preliminary (multi-view model) of our work and add UV visualization. In response, we have revised the manuscript to explicitly state the research problem (PBR texturing for a mesh) and added a new paragraph detailing the multi-view diffusion preliminaries, ensuring the introduction is more self-contained and accessible.

Reviewer RwEh (rating: 8) increased the confidence score to 5 after we clarified the novelty of our inpainting module.

---

**Novelty and Effectiveness**

Unlike prior multi-channel or two-stage approaches for PBR generation, we introduce a **novel multi-branch architecture** consisting of a **shaded branch** (providing illumination context for accurate PBR decomposition) and a **material branch**, connected through **lighting-aware attention**. With **multi-stage training and one-stage inference**, this design enables disentangled Albedo/MR reasoning and effectively mitigates (1) the domain gap between albedo and MR generation, (2) the severe data imbalance issue of albedo and MR in existing 3D datasets, and (3) achieves greatly improved physical plausibility. The experiments (Table 1) show that our method achieves ~35% improvement over SoTA open-source baselines on key metrics, and ~20% improvement over methods trained on private datasets. Moreover, unlike previous inpainting modules that operate directly in UV space, we are the first to explore a 2D foundation model for texture inpainting tasks. This alleviates UV ambiguity and produces view-consistent, seamless completions across occluded regions (Fig. 7).

**Experiment Cases**

To ensure a direct and fair evaluation with existing baselines, some of the input cases (i.e., reference images) are consistent with the prior works in the main paper, as also noted by the open reviewer. This demonstrates that our method performs robustly even without careful case selection, further supporting the reliability and generality of our approach.

Best regards,

Authors of Submission 1297

---

### Meta-Review · Area_Chair_WZaP · 2026-01-06

**Summary:**

Following is a summary of the reviewers' major concerns:

### Reviewer t6H4
1. The idea of using lighting priors for material generation is not novel.
    * **Author replies**: Effectively integrating lighting priors is non-trivial.  The paper introduces a novel multi-branch design with an implicit lighting-aware attention mechanism.
    * **AC comments**: I think the concern is well addressed.
2. The method is computationally intensive, and the resolution is limited to 768x768 and cannot go up to 2K/4K.
    * **Author replies**: Training time does not limit real-time or interactive applications.
    * **AC comments**: I think the concern is well addressed.
3. Lack of support for complex materials such as reflective materials and subsurface scattering.
    * **Author replies**: This is out of the scope of the paper. Previous works also do not support this.
    * **AC comments**: I think the concern is well addressed.
4. Heavily relies on Objaverse and generalization to OOD input seems to be limited.
    * **Author replies**: The paper has included extensive OOD data across both input meshes and reference images.
    * **AC comments**: I think the concern is well addressed.
5. Results under extreme lighting scenarios, such as strong backlighting, are not well documented.
    * **Author replies**: The authors added additional results on highly reflective materials and strong backlighting conditions.
    * **AC comments**: I think the concern is well addressed.



### Reviewer yPWH
1. The main method of the paper is not well presented, and it's difficult to follow the paper.
    * **Author replies**: The authors make further clarification on the paper
    * **AC comments**: I share the same concern. I believe that the writing of the paper needs to be
    significantly improved to make it easier to follow.
2. Actual PBR materials need to be presented instead of shaded versions.
    * **Author replies** The authors added PBR textures (UV) to the paper.
    * **AC comments**: I think the concerns are well addressed.
3. Shading is baked in albedo.
    * **Author replies**: The authors acknowledge the limitation, but argue that the proposed method
    performs the best when compared to baseline methods.
    * **AC comments**: I think it's a valid concern, but considering that it's a challenging problem
    and the proposed method has achieved better performance, I believe the concern is well addressed.


### Reviewer RwEh
* No major concerns.

### Reviewer EhkN
* No major concerns.

**Reviewer Concerns:**

See above.

I think the major outstanding concern is the presentation of the paper. The paper needs to be revised to make each component clearer. Particularly, how each component is trained, what the input and output are, and what the loss function is. However, I do think this concern can be addressed in the final version.

**Reviewer Scores:**

Reviewer t6H4: I think the reviewer would raise the score to 4 or 6.

Reviewer yPWH: I think the reviewer would raise the score to 4 or 6.

Reviewer RwEh: maintain the score.

Reviewer EhkN: maintain the score.

---

### Decision · Program_Chairs · 2026-01-26

Accept (Poster)